# The cytoplasm of living cells can sustain transient and steady intracellular pressure gradients

**Majid Malboubi[1,2]\*, Mohammad Hadi Esteki[3], Malti B Vaghela[2], Lulu IT Korsak[4], Ryan J Petrie[4], Emad Moeendarbary[3], Guillaume Charras[2,5,6]\***

[1]Department of Mechanical Engineering, University of Birmingham, Birmingham, United Kingdom; [2]London Centre for Nanotechnology, University College London, London, United Kingdom; [3]Department of Mechanical Engineering, University College London, London, United Kingdom; [4]Department of Biology, Drexel University, Philadelphia, United States; [5]Institute for the Physics of Living Systems, University College London, London, United Kingdom; [6]Department of Cell and Developmental Biology, University College London, London, United Kingdom

## eLife Assessment

This **important** study combines imaginative and innovative experiments with a finite element modelling to demonstrate the relevance of poroelasticity in the mechanical properties of cells across physiologically relevant time and length scales. The authors present **convincing** evidence that cytosolic flows and pressure gradients can persist in cells with permeable membranes, generating spatially segregated influx and outflux zones. These findings are of interest to the cell biology and biophysics communities.

\*For correspondence:
m.malboubi@bham.ac.uk (MM);
g.charras@ucl.ac.uk (GC)

Competing interest: The authors declare that no competing interests exist.

**Abstract** Understanding the physical basis of cellular shape change in response to both internal and external mechanical stresses requires characterisation of cytoplasmic rheology. At subsecond time-scales and micron length-scales, cells behave as fluid-filled sponges in which shape changes necessitate intracellular fluid redistribution. However, whether these cytoplasmic poroelastic properties play an important role in cellular mechanical response over length- and time-scales relevant to cell physiology remains unclear. Here, we investigated whether and how a localised deformation of the cell surface gives rise to transient intracellular flows spanning several microns and lasting seconds. Next, we showed that pressure gradients induced in the cytoplasm can be sustained over several minutes. We found that stable pressure gradients can arise from the combination of cortical tension, cytoplasmic poroelasticity, and water flows across the membrane. Overall our data indicate that intracellular cytosolic flows and pressure gradients may play a much greater role than currently appreciated, acting over time- and length-scales relevant to mechanotransduction and cell migration, signifying that poroelastic properties need to be accounted for in models of the cell.

## Introduction

The rheology of cells determines their response to internal and external mechanical stimuli encountered during normal physiological function. Since the cytoplasm forms the largest part of the cell by volume, its rheological properties are key to understanding cellular shape change in response to mechanical stress. The cytoplasm can be described as a biphasic material that consists of a porous

solid phase bathed in an interstitial fluid (*Moeendarbary et al., 2013*). In such a material, stress relaxation arises from fluid flow through the pores of the solid phase as a result of spatial gradients in the applied stress field (*Biot, 1941*). By studying stress relaxation in microindentation experiments combined with perturbations including volumetric deformations together with chemical and genetic treatments, previous work has shown that, at subsecond time-scales and micron length-scales, the cytoplasm behaves as a poroelastic material (*Moeendarbary et al., 2013*; *Esteki et al., 2021*). However, the contribution of cytoplasmic poroelastic properties to the mechanical response of cells has not been characterised on the tens of micron length-scale and second-to-minute time-scale relevant to cell physiology. Thus, it is unclear if poroelastic properties must be accounted for in models of cell mechanics.

During mechanotransduction, cells sense external mechanical stresses and translate them into biochemical signals (*Hoffman and Crocker, 2009*). The intracellular fluid flows and pressure gradients elicited in poroelastic materials in response to local application of stress may represent a stimulus for detection. While deformations are applied at the cellular scale, most mechanosensory processes act at the molecular scale through opening of ion channels and unfolding of proteins. Thus, the spatial extent of the stress and strain fields will determine what proportion of the cell is likely to respond to a stimulus and the temporal evolution of these stress and strain fields will determine the duration over which mechanosensory processes will be stimulated. Understanding how stresses equilibrate in the cytoplasm in response to local application of force is necessary to better understand the physical parameters detected by intracellular mechanotransductory pathways. However, direct characterisation of the spatiotemporal deformation induced by sudden local application of force is currently lacking.

Migrating cells often adopt a polarised morphology with a gradient of non-muscle myosin II (NMII) protein increasing towards the rear (*Bergert et al., 2015*; *Liu et al., 2015*; *Ruprecht et al., 2015*) or the front (*Newman et al., 2023*; *Petrie et al., 2014*), depending on cell type. Given the poroelastic nature of cytoplasm, theoretical considerations predict that such cortical myosin gradients should result in intracellular pressure gradients driving intracellular fluid flows in the direction opposite to the gradient of NMII (*Hawkins et al., 2011*; *Taber et al., 2011*). Intracellular flows have been proposed to participate in protrusion formation and migration but have only been indirectly observed in cells (*Keren et al., 2009*; *Zicha et al., 2003*; *Iwasaki and Wang, 2008*; *Loitto et al., 2009*; *Manoussaki et al., 2015*; *Stroka et al., 2014*). These flows may play a particularly important role during migration in confined environments with low adhesion, where most cell types adopt an amoeboid morphology and extend pressure-driven protrusions at their front (*Bergert et al., 2015*; *Liu et al., 2015*; *Ruprecht et al., 2015*; *Bergert et al., 2012*; *Tyson et al., 2014*; *Wilson et al., 2013*). Although significant evidence points towards a role for myosin-generated pressure gradients in migration (*Petrie et al., 2014*), it is unclear if intracellular pressure gradients can be sustained over the minute-long time-scales involved in migration.

Here, we used a combination of cell physiology experiments, high-resolution nanoparticle tracking, and finite element (FE) simulations to study how the combination of membrane permeability and cytoplasmic poroelastic properties can lead to steady-state gradients in intracellular pressure. We investigate the dynamics of cellular stress relaxation in response to external and internal mechanical stresses to determine if the poroelastic nature of cytoplasm is relevant to cell physiology on tens of microns length-scales and minute time-scales. We observe a cell-scale mechanical response following application of a local deformation to the cell surface and show that it is due to transient intracellular fluid flows elicited in poroelastic materials. We then reveal experimentally that intracellular pressure gradients lasting several minutes can be sustained in the cytoplasm, signifying that pressure gradients may play a role in migration.

## Results and discussion
### Whole-cell mechanical equilibration in response to localised deformation necessitates several seconds

Cells are often subjected to localised mechanical forces applied at high strain rates (*Avril et al., 2011*; *Li et al., 2011*; *Perlman and Bhattacharya, 1985*). In poroelastic materials, rapid application of localised stress pressurises the interstitial fluid. Stress relaxes by flow of water out of the deformed

region through the pores of the solid phase. As a consequence, the rate of mechanical equilibration is set by the poroelastic diffusion constant $D_p$ that depends on the elasticity $E$ of the solid phase, the viscosity $\eta$ of the fluid phase, and the size $\xi$ of the pores through which the interstitial fluid can permeate: $D_p \sim E\xi^2/\eta$ (**Charras et al., 2005**). The time-scale of water flow out of the deformed region is $t_p \sim L^2/D_p$, where $L$ is a length-scale associated with deformation. Previous work has shown that local force application to a cell can result in global changes in the height of the cell surface and revealed the presence of two regimes: one fast due to stress propagation in the cytoskeleton and the other slow, whose origin was unclear (**Rosenbluth et al., 2008**). As previous work has shown that the cytoplasm behaves as a poroelastic material (**Moeendarbary et al., 2013**), we hypothesised that the slow response taking place of over tens of microns length-scale and seconds time-scale reflects the poroelastic nature of the cytoplasm.

To examine the role of poroelasticity in the mechanical response of cells, we locally deformed the cell surface with the tip of an AFM cantilever. In our experiments, the deformation had an estimated characteristic length $L \sim 3$ µm ($L \sim \sqrt{d\delta}$ with $d \sim 4$ µm the diameter of the indentation and $\delta \sim 2$ µm the indentation depth) applied with a rise time $t_r \sim 100$ ms (**Figure 1A, B**). In HeLa cells, previous work has reported $D_p \sim 40$ µm²/s (**Moeendarbary et al., 2013**), leading to an estimate of the characteristic time of fluid efflux $t_p \sim 200$ ms. As $t_p$ is larger than the rise time $t_r$, poroelasticity may contribute to mechanical relaxation.

To detect the global response of the cell surface to local force application with high spatial accuracy, we employed defocusing microscopy (**Esteki et al., 2021**; **Rosenbluth et al., 2008**). In this technique, collagen-coated fluorescent beads are tethered to the cell surface. The plane of focus is chosen such that the beads are deliberately out of focus and display multiple diffraction rings about their centre (**Figure 1—figure supplement 1A**). Bead vertical displacement is monitored with nanometre precision by measuring the temporal evolution of the diameter of the outer diffraction ring in response to the deformation of the cell surface induced by AFM (**Figure 1—figure supplement 1A, B**). At steady-state, beads closer than 6 µm to the AFM tip moved downwards while beads further away moved upwards, as expected for an elastic material subjected to indentation and as previously observed (**Rosenbluth et al., 2008**; **Figure 1C**). The amplitude of the steady-state displacement decreased with distance between the bead and the AFM tip (**Figure 1C**, **Figure 1—figure supplement 3A**). The temporal evolution of the bead movement displayed a biphasic response at all positions throughout the cell (**Figure 1C**). The first phase consisted in a fast movement of all the beads independent of their distance from the AFM tip over a time-scale shorter than ~0.3 s, possibly due to cell surface tension (**Figure 1C**, inset, **Figure 1—figure supplement 2**). In the second phase, the beads relaxed to their final displacement over a characteristic time-scale $\tau_p$ that increased with increasing distance between the AFM tip and the bead (**Figure 1—figure supplement 2**, **Figure 1C–E**, Methods). The overall relaxation of the second phase lasted up to ~5 s. Thus, local application of external force leads to a whole-cell mechanical equilibration lasting several seconds, consistent with previous work (**Rosenbluth et al., 2008**).

When stress is applied to a poroelastic material, relaxation takes place through flow of fluid out of the pressurised region at a rate that depends on the hydraulic permeability of the cytoplasm and hence its pore size. If the slow relaxation observed in cells is indeed due to poroelastic effects driving cellular-scale intracellular fluid flows, changes in the hydraulic pore size $\xi$ should affect the duration of the second phase of bead movement but not the first phase. Pore size can be decreased by increasing the osmolarity of the medium (**Moeendarbary et al., 2013**; **Esteki et al., 2021**). Increasing osmolarity by 300 mOsm led to a ~3.5-fold increase in the median characteristic equilibration time $\tau_p$ of the slow phase (**Figure 1D**). The amplitude of the first phase of bead movement was unaffected (**Figure 1—figure supplement 3B**) and we were unable to determine changes in the duration of the first phase because of its short duration. Conversely, when we treated cells with latrunculin, $\tau_p$ showed a trend towards decrease (**Figure 1D**), consistent with previous reports showing that $D_p$ increases with latrunculin treatment (**Moeendarbary et al., 2013**). Surprisingly, the amplitude of bead movement was not affected (**Figure 1—figure supplement 3**). Thus, the second-long global changes in cell surface height that occur in response to local application of stress are qualitatively consistent with a poroelastic cytoplasm.

## Cytoplasmic flows induced by microinjection follow Darcy's law

In porous media, pressure imbalances are dissipated by interstitial fluid flows. Though intracellular fluid flows have been inferred to be driven by internal stresses during cell motility (**Hawkins et al.,**

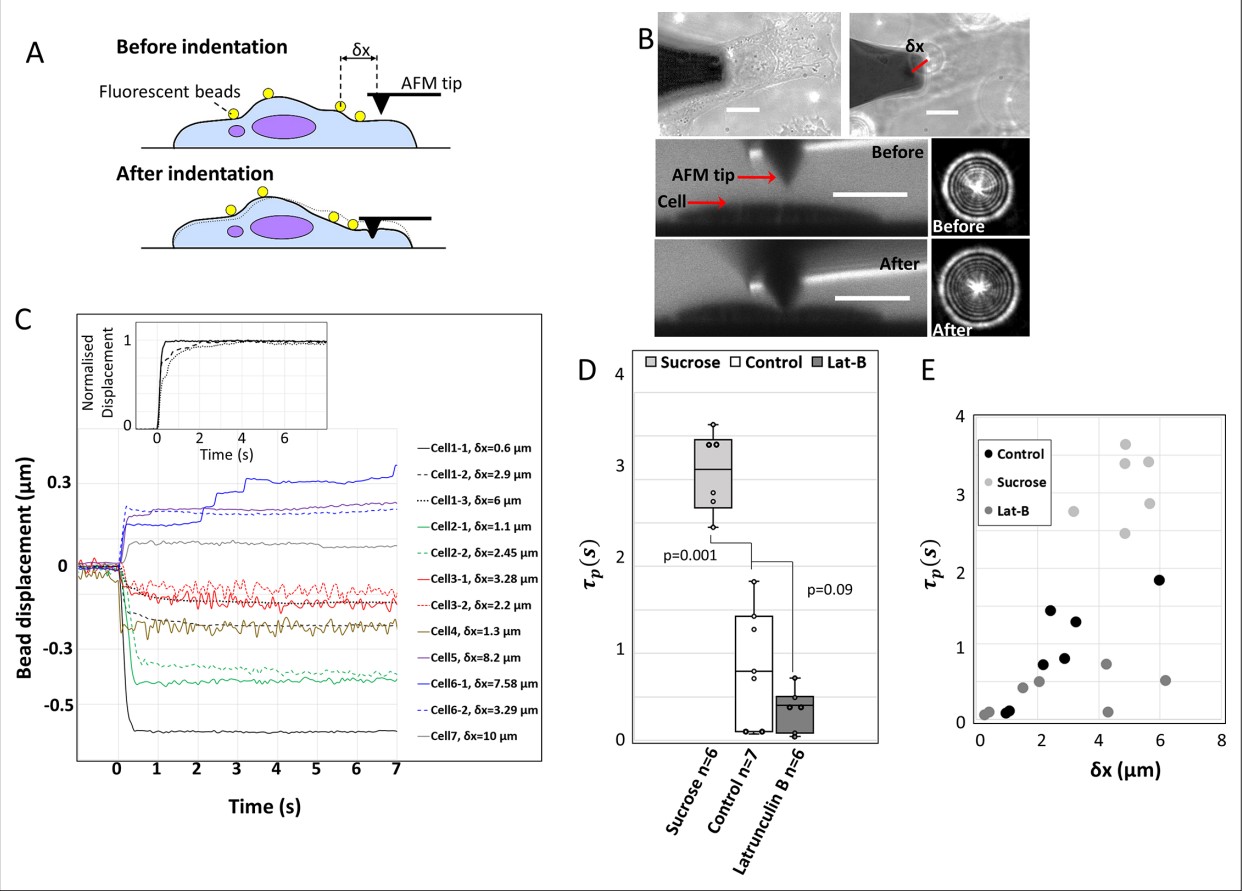

**Figure 1.** Local application of stress gives rise to cell-scale intracellular flow. (**A**) Schematic diagram of the experiment. Collagen-coated fluorescent beads are bound to the cell surface. An AFM cantilever is brought into contact with one side of the cell and bead movement in the z-direction is imaged over time. (**B**) Representative image showing a combined phase contrast and fluorescence image of the beads on the cell. Top panels: left: The AFM cantilever appears as a dark shadow on the left of the image. The bead is visualised by fluorescence. When the plane of focus is moved higher than the cell, a halo of fluorescence centred on the bead appears (middle panel, right). The diameter of the halo of the bead reports on the distance between the bead and the plane of focus (see Methods, *Figure 1—figure supplement 1A*). Variations in halo radius indicate changes in height caused by indentation. The distance between the bead and the AFM tip is indicated by a red line. Middle panels: left: Profile of a cell before indentation. A cell-impermeable fluorescent indicator has been added to the medium and the cell appears dark. The AFM cantilever was imaged by reflectance and appears bright. Right: Representative image of the halo before indentation. Bottom panels: left: Profile of the same cell as in the middle panel during indentation by an AFM cantilever. Right: Halo of the same bead as in the middle panel during indentation. Scale bar = 10 µm. (**C**) Change in bead height as a function of time for a total of 12 beads on 7 cells. Beads from the same cell appear in the same colour. Inset shows normalised displacement of three beads on the same cell located at different distances from the AFM tip. This highlights the slower response in the second phase for more distant beads. The colour code in the inset is the same as the main figure. (**D**) Characteristic relaxation time $\tau_p$ of the second phase for control cells (n = 7 cells) and cells treated with sucrose (n = 6 cells) and latrunculin (n = 7 cells). In the box plot, the black line is the median, the bottom and top edges of the box indicate the 25th and 75th percentiles, respectively. The whiskers extend to the most extreme data points that are not outliers. Data points appear as black dots. Conditions were compared with a Wilcoxon rank sum test. (**E**) Characteristic relaxation time $\tau_p$ as a function of distance from the AFM tip for control cells (black), sucrose (light grey), and latrunculin (dark grey).

The online version of this article includes the following figure supplement(s) for figure 1:

**Figure supplement 1.** Experimental setup.

**Figure supplement 2.** Definition of $\tau_p$ in AFM experiments.

**Figure supplement 3.** Surface movement in response to application of extrinsic force by AFM indentation.

*2011*; *Taber et al., 2011*; *Keren et al., 2009*; *Iwasaki and Wang, 2008*; *Herant et al., 2003*) or external stresses during stress relaxation (*Rosenbluth et al., 2008*; *Moeendarbary et al., 2013*) a direct evaluation of the relationship between the pressure gradient $\nabla P_f$ and the velocity of intracellular fluid flow $v$ is lacking in cells. In classic porous media, the two are linked by Darcy's law:

1. $v = -k\nabla P_f$ with $k$ the hydraulic permeability of the medium.

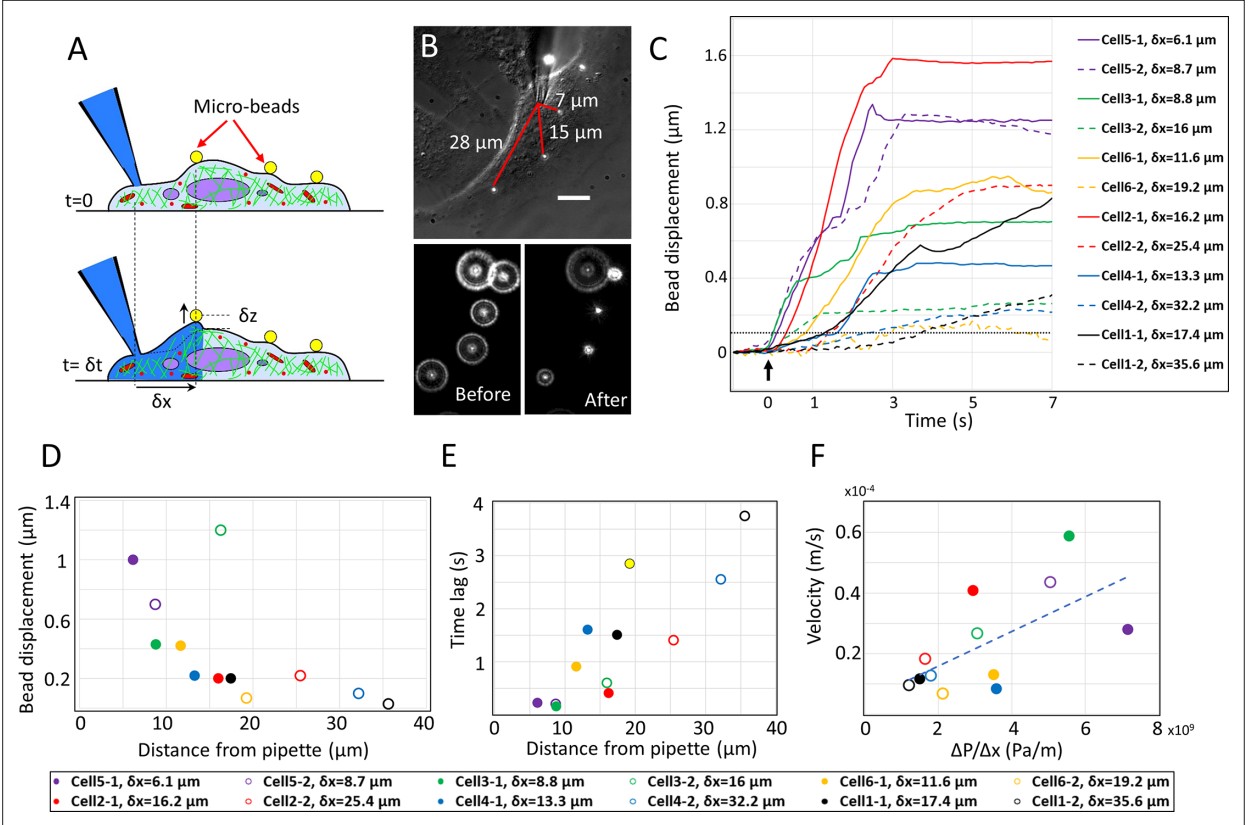

**Figure 2.** Intracellular fluid propagation in response to pressure gradients. (**A**) Schematic diagram of the experiment. Collagen-coated fluorescent beads (yellow) are bound to the surface of a cell in fluidic communication with a micropipette. At time $t = 0$ s, pressure is increased within the pipette leading to injection of fluid into the cell (dark blue). After a time-lag $\delta t$, the fluid propagation front reaches a bead in $\delta x$ resulting in an increase of its height by $\delta z$. (**B**) Top: Representative image showing a combined DIC and fluorescence image of a cell. The micropipette appears at the top of the image and a fluidic connection is established. The distance from the pipette tip to each bead is indicated by red lines. Scale bar = 10 μm. Bottom: Defocused image of the fluorescent beads tethered to the cell surface before (left) and after (right) propagation of the fluid flow through the cell. (**C**) Temporal evolution of height for beads situated at different distances from the pipette tip. Data from $N = 12$ beads from $n = 6$ cells. The distance of each bead is listed in the inset. Beads from the same cell are in the same colour. The time at which microinjection starts is $t=0$ s (indicated by the arrow). The beads respond to injection with a time lag that increases with increasing bead-pipette distance. The dashed black line corresponds to a 0.1 μm displacement of the beads from their initial position. This threshold is used to calculate the time-lag $\delta t$ between injection and bead response (see SI methods). (**D**) Bead displacement as a function of distance from the pipette after $t = 2$ s. (**E**) Time-lag $\delta t$ as a function of distance from the pipette. (**F**) Velocity as a function of estimated pressure gradient. (**D–F**) Beads from the same cell appear as solid or open markers of the same colour. Colour code is indicated on the right of panel **F**.

The online version of this article includes the following figure supplement(s) for figure 2:

**Figure supplement 1.** Fluid injection experiments and experimental setup.

To characterise intracellular flows in response to pressure gradients, we created a fluidic link between a micropipette and an interphase cell using techniques developed for electrophysiology (whole-cell recording) and applied a step pressure increase to the micropipette (*Figure 2A*, *Figure 2—figure supplement 1A*, Methods). In this configuration, the applied pressure results in fluid injection into the cell, causing the solid phase of the cytoplasm to expand and the height of the cell surface to increase as the fluid progressively permeates through the solid phase of the cytoplasm. In our experiments, we recorded bead displacement using defocusing microscopy for beads at different distances away from the micropipette (*Figure 2B, C*). After application of a pressure step, beads moved upwards with an amplitude that decreased with increasing distance from the micropipette for a given time (*Figure 2D*) and with a time lag that increased linearly with the distance between the bead and the micropipette (see Methods and *Figure 2E*), indicating an approximately constant velocity of the propagation front. When fluid injection was continued over longer time-scales (>5 s), cellular volume

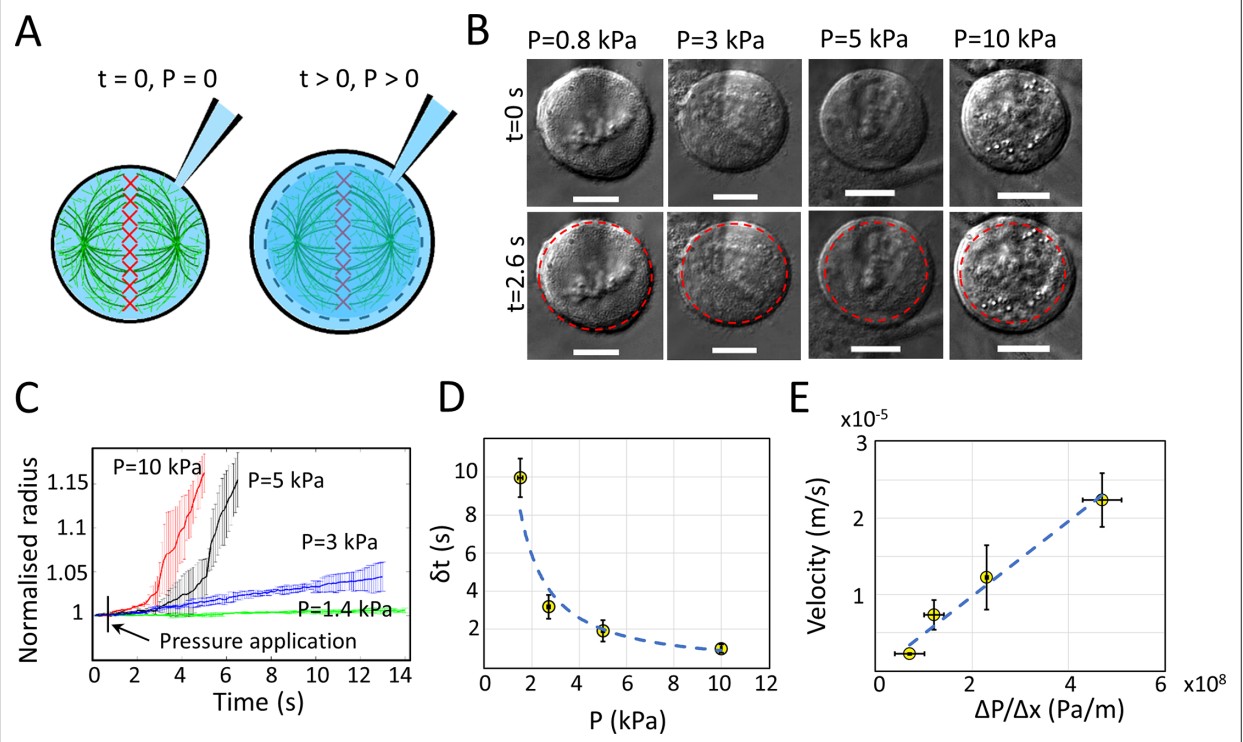

**Figure 3.** The cytoplasm of metaphase cells displays a porous behaviour. (**A**) Schematic diagram of the experiment examining the change in diameter of metaphase HeLa cells due to application of a step increase in pressure through the micropipette. To detect the arrival of fluid flow at the cell periphery following application of a step pressure in the micropipette, we monitored changes in cell diameter. An increase in cell diameter was detectable after a time delay $\delta t$ that depended on the amplitude of the pressure step. (**B**) DIC images of representative experiments. The step change δn pressure was applied at $t = 0$ s. Top: equatorial plane of cells blocked in metaphase at $t = 0$ s. The micropipette tip is out of the plane of focus. Bottom: Cells at $t = 2.6$ s after pressure application. The initial diameter is indicated by the red dashed circle. Scale bar = 10 μm. (**C**) Temporal evolution of the relative diameter of metaphase HeLa cells subjected to different pressure steps. Error bars represent standard deviations. $N = 3$ cells per pressure. The timing of pressure application is indicated by a vertical black line. The pressure corresponding to each curve is indicated on the graph. (**D**) Time-lag $\delta t$ between pressure application and the onset of diameter increase as a function of the amplitude of the pressure step. The dashed line indicates a hyperbolic fit. Whiskers indicate the standard deviation. $n = 3$ cells per pressure. (**E**) Velocity as a function of estimated pressure gradient. The dashed line indicates a linear fit. Whiskers indicate the standard deviation. $N = 3$ experiments per data point.

regulation mechanisms (**Hoffmann et al., 2009**) could not compensate for the volume increase and the membrane delaminated from the cortex forming large blebs before the cell eventually lysed.

In our experiments, the duration of microinjection (~2 s) is short compared to the characteristic time-scale of water flows across the cell membrane (~20–100 s) (**Potma et al., 2001**). Therefore, we can neglect fluid losses through the membrane and we expect flows to follow Darcy's law in response to microinjection. As the velocity of Darcy flows depends on the hydraulic permeability $k$ and the pressure gradient $\nabla P_f$, we plotted the estimated velocity $v \sim \frac{\Delta x}{\delta t}$ as a function of an estimate of the pressure gradient $\Delta P/\Delta x$, with $\Delta x$ the position of the bead relative to the micropipette and $\delta t$ the time lag of its movement relative to the onset of injection (see SI for estimation of injection pressure). This revealed a linear dependency with a slope $k \sim 1.25 \; 10^{-13}$ m²/(Pa s) (**Figure 2F**, $r^2 = 0.70$), close to previous estimates (**Moeendarbary et al., 2013**; **Charras et al., 2009**). We then repeated these experiments in metaphase cells in which we could precisely monitor diameter evolution after pressure application (**Figure 3A**). This revealed a strong dependency of the rate of change in diameter $d$ and the time-lag $\delta t$ (the onset of diameter increase) with applied pressure (**Figure 3B–D**). When we plotted an estimate of the velocity $v \sim \frac{d}{\delta t}$ as a function of an estimate of the pressure gradient $\nabla P_f \sim \Delta P/d$, this graph again revealed a linear relationship, as expected from Darcy's law, and yielded an estimate of $k \sim 5 \; 10^{-14}$ m²/(Pa s) (**Figure 3E**).

Together, these experiments show that pressure gradients lead to intracellular flows whose propagation follows Darcy's law with a hydraulic permeability consistent with those estimated by other experimental approaches (**Moeendarbary et al., 2013**; **Esteki et al., 2021**; **Charras et al., 2009**).

Counterintuitively, the hydraulic permeability in interphase cells was higher than in metaphase cells despite cell volume increasing in mitosis. This may be due to the profound remodelling of cytoplasmic organisation accompanying this stage of the cell cycle.

## Global cellular response to microindendation is compatible with a poroelastic cytoplasm

Next, we sought to gain insight into how the global cellular response to local indentation might arise from interplay of cellular poroelastic properties with other cellular mechanical properties. Previous work has hypothesised that local application of mechanical stress leads to a localised outflow of interstitial fluid from the deformed region that can then propagate through the cell (*Moeendarbary et al., 2013*; *Rosenbluth et al., 2008*). If vertical displacements in indentation experiments were just due to fluid propagation, we would expect that the onset of displacement would occur with a time lag that depends on distance as in the microinjection experiments (*Figure 2C, D*). Yet, the first phase of displacement shows no time lag and vertical displacements of the cell surface are observed far from the point of indentation (*Figure 1C*, inset). This indicates that poroelastic properties alone cannot explain the first phase of displacement.

One potential mechanical origin for the first phase may be the submembranous cortex whose mechanics differ markedly from the cytoplasm (*Vargas-Pinto et al., 2013*) and which generates a surface tension $\gamma$ through myosin contractility (*Salbreux et al., 2012*). The relative importance of surface tension in the cortex compared to elastic restoring forces in the cytoplasm can be grasped from the length-scale $l \sim \frac{\gamma}{E}$ with $E$ the elasticity of the cytoplasm. For length-scales larger than $l$, the effect of surface tension is negligible and elastic restoring forces are dominant. Using characteristic values of $\gamma \sim 1$ mN/m (*Chugh et al., 2017*) and $E_{cytoplasm} \sim 100$ Pa (*Moeendarbary et al., 2013*), we obtain a length-scale of ~10 μm, consistent with our experiments (*Figure 1D*). Furthermore, this scaling may explain why latrunculin treatment does not perturb the amplitude of bead movement in the first phase (*Figure 1—figure supplement 3B*). Indeed, latrunculin affects both the surface tension and the elastic modulus, potentially leading to compensation.

To determine if surface tension is important in setting the scale of the movement of the cell surface, we examined deformation of the membrane induced by indentation of the cell by a sharp AFM tip using mitotic cells to isolate the role of the cortex from cell shape and adhesion (*Figure 4—figure supplement 1*). The surface profile was imaged before and during a 2 μm depth indentation and, after segmentation, we quantified the distance from the tip at which the membrane deformation reached half of the maximum indentation depth (*Figure 4—figure supplement 1C*). In control conditions, the distance of the half maximum depth was 1.2 ± 0.2 μm and this decreased significantly to 0.8 ± 0.4 μm when surface tension was decreased with blebbistatin (*Figure 4—figure supplement 1D*). Therefore, surface tension participates in setting the length-scale of cell surface deformation in response to localised indentation.

The fast vertical displacement is followed by a second slow phase of displacement. With our experimental estimates of $k$, we can estimate the time-scale of relaxation of a vertical displacement arising from a local deformation of a tensed membrane tethered to a poroelastic cytoplasm. For each position within the region affected by the local deformation either directly or indirectly via surface tension, the time-scale for relaxation is $\tau_z \sim L^2/D_p$. $D_p \sim kE$ yielding a numerical estimate of $D_p \sim 10^{-11}$ m²/s and $L^2 \sim l\delta_z$ with $l \sim 10$ μm the distance over which vertical displacements are observed and a displacement amplitude $\delta_z \sim 0.5$ μm. With these values, we find $\tau_z \sim 500$ ms, qualitatively consistent with the experimentally observed times (*Figure 1D*).

Thus, the instantaneous displacement of the cell surface far from the region of indentation may be due to cellular surface tension (*Figure 1C*) and these vertical displacements may drive intracellular fluid flows throughout the poroelastic cytoplasm.

## Cytoplasmic poroelasticity combined with membrane permeability allows formation of stable intracellular pressure gradients

External stresses applied to the cell surface give rise to transient intracellular pressure gradients that are dissipated by molecular turnover and intracellular flows. However, spatial inhomogeneities in internal stresses generated by the actomyosin cytoskeleton can be maintained for tens of minutes because they arise as a natural consequence of molecular turnover and contractility (*Prost et al., 2015*). For

example, during migration, the presence of gradients in myosin motors increasing from front to rear suggests the existence of a high cortical tension at the cell rear that can be sustained without relaxing (*Hawkins et al., 2011*; *Taber et al., 2011*). One implication, given the poroelastic properties of the cytoplasm, is that this surface gradient should result in a sustained pressure gradient continuously driving intracellular fluid flows towards the cell front. Cytoplasmic pressure gradients have been observed in fibroblasts migrating on 2D substrates (*Iwasaki and Wang, 2008*) and in 3D matrices (*Petrie et al., 2014*; *Petrie et al., 2017*). In addition, intracellular fluid flows have been inferred in cells migrating on 2D substrates (*Keren et al., 2009*; *Iwasaki and Wang, 2008*) and suggested to play a motive role in cells migrating through confined environments (*Stroka et al., 2014*).

Because this phenomenon involves a complex interplay between surface tension generated by the cortex, poroelastic behaviour of the cytoplasm, and fluid permeation through the membrane, deriving analytical solutions is challenging and we therefore turned to FE modelling to gain a qualitative understanding of how long-lasting pressure gradients can be maintained in the cell. In our model, we modelled the cell as a pressurised poroelastic material surrounded by a less permeable surface region 250 nm in thickness, representing the membrane and the cortex (*Potma et al., 2001*). For this, we parameterised a poroelastic FE model (*Figure 4A, B*, Methods), adjusting the values of $E$ and $D_p$ such that our model could predict the cell's response to a localised deformation (*Figure 4C*) and the dynamics of displacement of the cell surface in response to microinjection (*Figure 4D, E*).

After parameterising our model, we then introduced a sink in a region of the cell periphery (2 μm in diameter) to simulate a region of lower pressure (*Figure 4F*). In this region, fluid can rapidly leave the cell, which leads to a reduction in cell volume. In turn, this volume reduction will increase intracellular osmolarity and drive water influx across the membrane in the rest of the cell surface. If the membrane permeability is large enough, efflux through the sink can be compensated by influx through the membrane to maintain a constant cell volume.

When we computationally applied a suction pressure to the sink region, our model predicted a spatial gradient of intracellular pressure with a low pressure close to the sink that increased towards the initial cell pressure far away from this region (*Figure 4G–I*). To gain insights into the importance of the membrane–cortex layer for generation of this pressure gradient, we varied cortical thickness and diffusion constant in our model. While cortex thickness had little influence (*Esteki et al., 2021*), diffusion through the cortex strongly affected the length-scale of the gradient (*Figure 4J*), with high diffusion constants leading to gradients that reached the initial cell pressure over short length-scales. This is consistent with previous work that showed that localised exposure of cells to high osmolarity medium leads to localised dehydration with a sharp transition in pore size (and hence $D_p$) between the exposed and non-exposed region (*Charras et al., 2009*). This work suggested that membrane permeability was on the order of 100 times lower than cytoplasmic permeability, within the lower range of the parameter values tested in our model (red curve, *Figure 4J*). Thus, cytoplasmic poroelasticity combined with passage of water across the membrane can in principle allow maintenance of stable intracellular pressure gradients and pressure compartmentalisation in living cells.

## Cells can accommodate intracellular pressure gradients over minute time-scales

To test our predictions experimentally, we introduced a local depressurisation on the cell surface by establishing a fluidic link using a micropipette and bringing its back-pressure to atmospheric pressure. We verified that, in these conditions, a small suction was generated at the tip of the micropipette (*Figure 4—figure supplement 2*), signifying that the pressure applied at the pipette tip was lower than the intracellular pressure. Based on the cellular poroelastic properties, the pipette dimensions, and the cell dimensions, steady state is reached for $t_p \sim \frac{R_{cell}^2}{D_p} \sim 2.5\text{s}$ after pressure release, assuming that relaxation is entirely limited by cellular poroelastic properties.

We verified that, following establishment of a fluidic connection between the cell and the micropipette, no occlusion occurred over time and that actomyosin was not perturbed (*Figure 4—figure supplements 3 and 4*). To confirm fluid efflux from the cell into the micropipette, we labelled cells with a fluorescent dye that becomes cell-impermeant upon cleavage by cellular proteases. We compared the temporal evolution of fluorescence in pairs of cells, one subjected to depressurisation and a neighbour that was unperturbed (*Figure 5*). Fluorescence intensity remained constant in the control cells but decreased approximately linearly over the course of 10 min in the depressurised cells, indicating

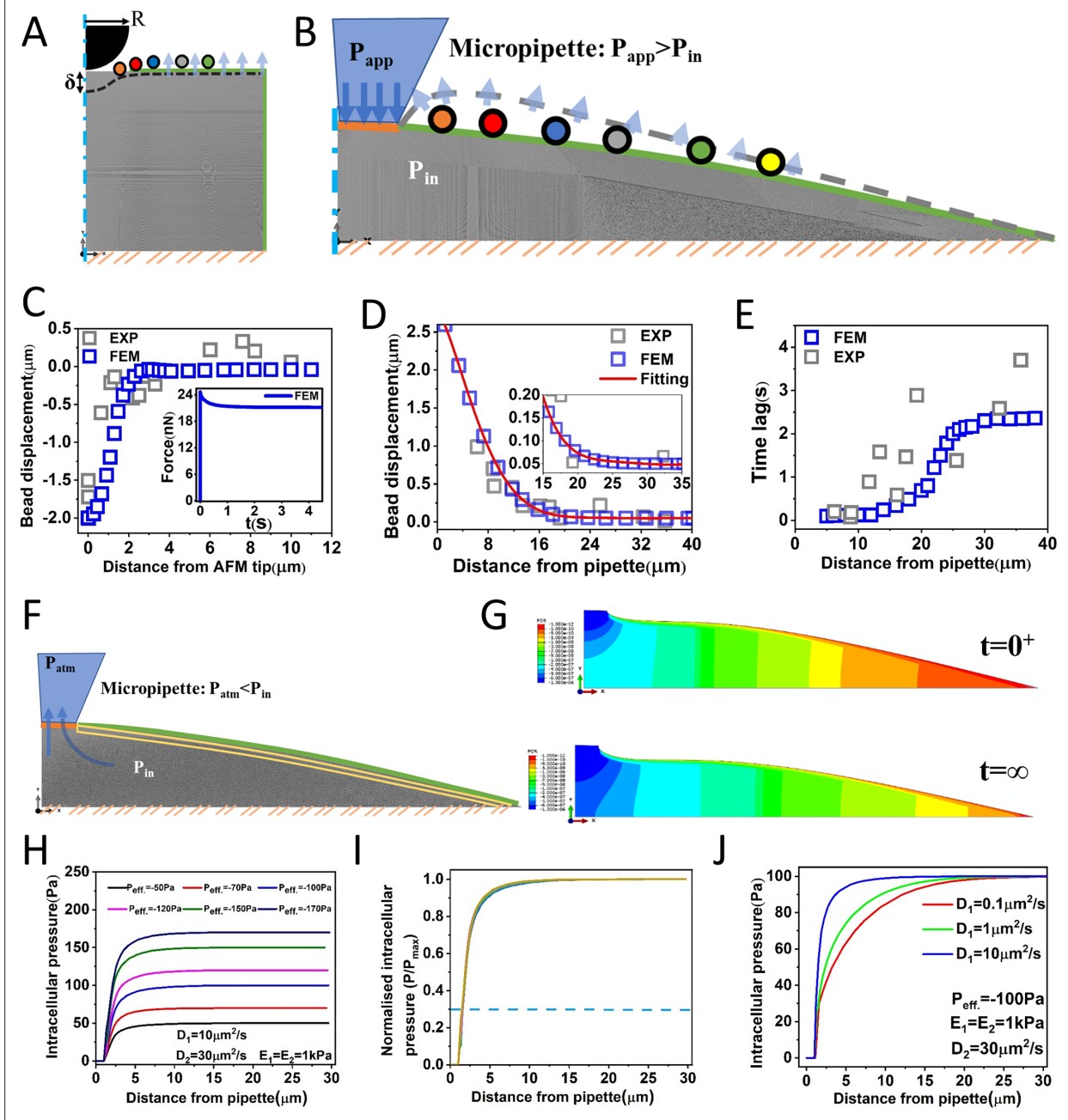

**Figure 4.** A poroelastic cytoplasm enables the emergence of steady-state pressure gradients. (**A**) Schematic representation of a quasi two-dimensional slab of poroelastic material with a deformable upper surface. The material is parameterised by its elasticity $E$, and its poroelastic diffusion constant $D$. The bottom surface is uniformly tethered to an impermeable infinitely stiff material. The top surface is subjected to an indentation $\delta$ applied by a spherical indenter of radius $R$. The surface profile after deformation is indicated by the dashed black line. (**B**) Schematic representation of a two-dimensional slab of poroelastic material (grey). The cytoplasmic material is parameterised by its elasticity $E$, its poroelastic diffusion constant $D$, and its pressure $P_{in}$. One part of the surface is permeable and fluid is injected into the cell through this region at a pressure $P_{app}$. The bottom surface is uniformly tethered to an impermeable infinitely stiff material. The surface profile after fluid injection is indicated by the dashed grey line. (**C**) Experimental (grey) and predicted steady-state vertical displacement of beads tethered to the cell surface in response to localised indentation. Inset shows the predicted force–relaxation as a function of time. (**D**) Vertical displacement of beads in response to fluid microinjection after 2 s as a function of distance from the micropipette. Grey data points indicate experiments and blue data points the simulation. The red line indicates the trendline. Inset shows a zoom on the further distances. (**E**) Time lags of the onset of vertical displacement as a function of distance from the micropipette. Grey data points indicate experiments and blue data points the simulation. (**F**) Schematic diagram of depressurisation experiment. A two-dimensional slab of poroelastic material (grey) surrounded by a less permeable, thin, outer layer (yellow) representing the cortex. The cytoplasmic material is parameterised by its elasticity $E_2$, its poroelastic diffusion constant $D_2$, and its pressure $P_{in}$. The outer layer is parameterised by $E_1$, $D_1$, and $P_{in}$. One part of the surface is permeable and fluid

*Figure 4 continued on next page*

*Figure 4 continued*

is released from the cell through this region to atmosphere pressure $P_{atm}$. At $t = 0$ s, the cell is subjected to a suction $P_{in}$ through the micropipette. (**G**) Pressure distribution immediately after depressurisation (top) and at steady state (bottom). (**H**) Intracellular pressure as a function of distance from the micropipette for a range of internal pressures $P_{in}$. (**I**) Intracellular pressure profiles from *H* normalised to the pressure at $x = 30$ µm from the micropipette. (**J**) Intracellular pressure distribution as a function of distance from the micropipette for different membrane–cortex poroelastic diffusion constants $D_1$.

The online version of this article includes the following figure supplement(s) for figure 4:

**Figure supplement 1.** The length-scale of surface deformations is controlled by cell surface tension.

**Figure supplement 2.** A positive pressure must be applied to the pipette to generate outflow.

**Figure supplement 3.** The pipette access resistance stays constant during whole-cell patch-clamp configuration.

**Figure supplement 4.** F-actin localisation at the interface between the cell and the micropipette during a pressure release experiment.

the presence of a constant pressure gradient and efflux from the cell. Finally, we asked if the cell volume remained constant during depressurisation by examining the change in radius of prometaphase cells subjected to depressurisation. In these cells, cell radius varied by less than 5% over 10 min of pressure release with no systematic trend to increase or decrease (*N* = 3 cells, *Figure 5—figure*

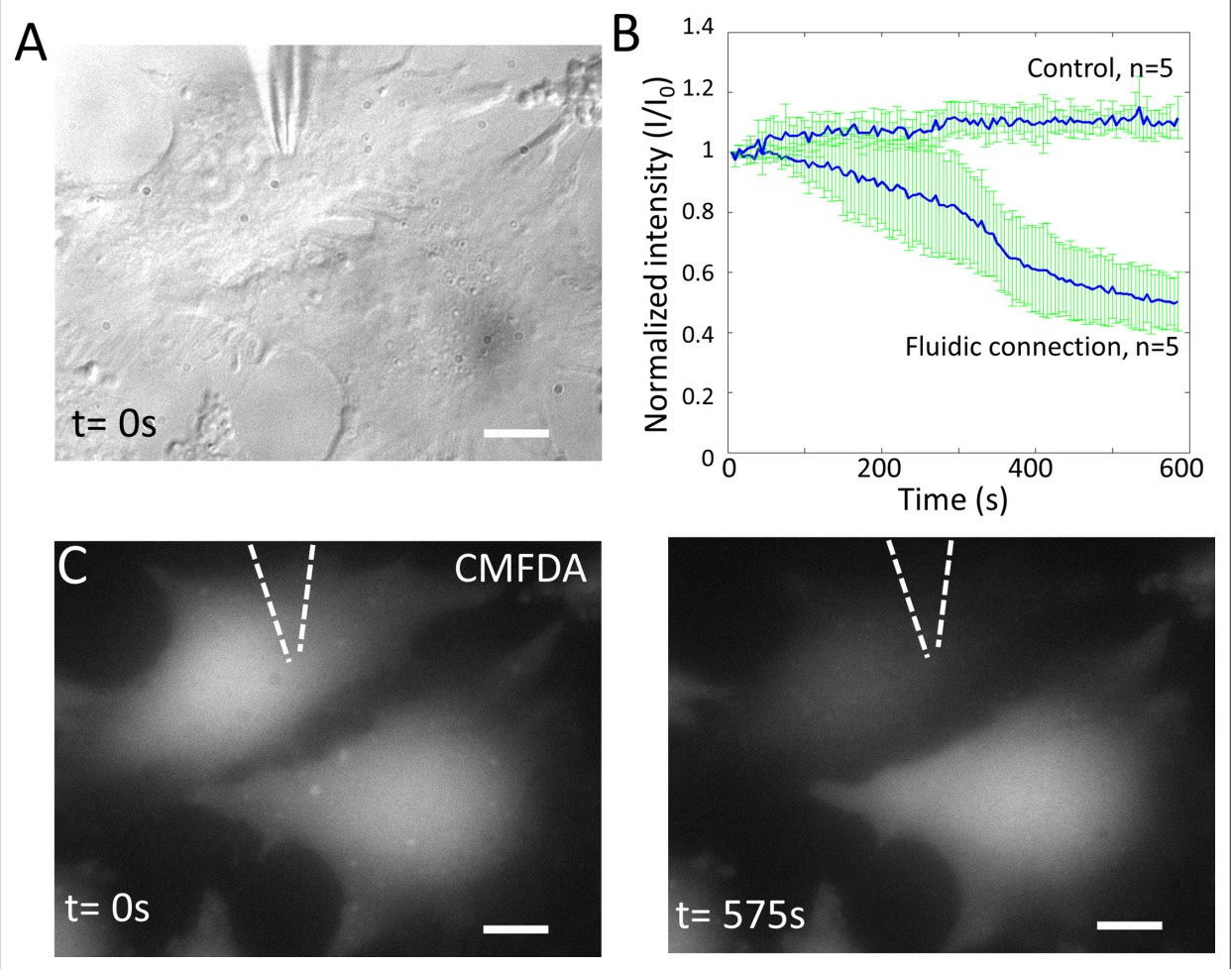

**Figure 5.** Whole-cell patch clamp and pressure release give rise to an outflow of fluid from the cell. (**A**) Differential interference contrast image of a typical experiment. The top cell is subjected to pressure release, while the bottom cell is not. Scale bar = 5 µm. (**B**) Temporal evolution of CMFDA cytoplasmic fluorescence intensity in control cells and cells subjected to depressurisation. The solid lines indicate the average and the whiskers the standard deviation. *N* = 5 experiments were averaged for each condition. (**C**) Fluorescence intensity of CMFDA in the cells in A prior to and after 575 s of depressurisation. Scale bar = 5 µm.

The online version of this article includes the following figure supplement(s) for figure 5:

**Figure supplement 1.** Cell volume remains constant during pressure release experiments.

*supplement 1*). Overall, these experiments indicate that cells have a sufficiently large membrane permeability to allow rapid exchange of fluid across the membrane to maintain a constant volume and that we can experimentally apply a long-lasting localised depressurisation.

Our simulation results indicate that, for values of $D_p$ measured in the cytoplasm, the low pressure region can be confined to a small zone near the pipette if membrane permeability is sufficiently large (*Figure 6C*). To experimentally test this, we used cell blebs as pressure gauges. Blebs are quasi-spherical protrusions of the cell membrane that arise due to pressurisation of the cytoplasm by cortical actomyosin contractility (*Charras et al., 2005*; *Tinevez et al., 2009*). First, we examined naturally occurring blebs in M2 melanoma cells (*Cunningham et al., 1992*). In control conditions, M2 cells bleb profusely with an intracellular pressure $P \sim 400$ Pa but, when actomyosin contractility is inhibited with a Rho-kinase inhibitor (Y27632), cells no longer bleb (*Charras et al., 2005*) and intracellular pressure drops to $P \sim 100$ Pa (*Figure 6A*, see methods for pressure measurement). Therefore, if $D_p$ is large and pressure is poorly compartmentalised, local depressurisation should lead to fast global decrease in intracellular pressure and blebbing should cease. In contrast, if $D_p$ is low and pressure is strongly compartmentalised, blebbing should continue unperturbed (*Booker and Carter, 1986*). In our experiments, following local pressure release (*Figure 6B*), M2 cells blebbed for several minutes, far longer than necessary for intracellular pressure to reach steady state or for significant exchange of fluid to take place (for $t > 6$ min, 19/19 cells still blebbed, *Figure 6D*). Furthermore, no spatially localised inhibition of blebbing could be noticed close to the pipette (*Figure 6—figure supplement 1*). These results suggest that pressure is highly compartmentalised in melanoma blebbing cells and that pressure gradients can be maintained over several minutes. Our pressure measurements indicate that blebs cease to emerge if the intracellular pressure drops below ~100 Pa (*Figure 6A*). Based on this and assuming a diffusion constant of $D_p \sim 0.1$ μm²/s in the membrane–cortex, our model predicts that pressure release would only induce a sufficiently large depressurisation to stop blebbing in a region ~1.5 μm away from the pipette tip (*Figure 6C*), consistent with the lack of spatial inhibition of blebbing we observe.

We then verified if HeLa cells could also compartmentalise pressure. In these cells, blebs can be induced within ~2 min by partial depolymerisation of the F-actin cytoskeleton induced by treatment with low doses of latrunculin. In these conditions, blebs emerge because HeLa cells have an intracellular pressure ranging from ~400 Pa in interphase to ~600 Pa in metaphase (*Figure 6A*) and because, minutes after latrunculin treatment, cortical actomyosin structures still remain well-defined (*Figure 6—figure supplement 2*). When contractility is blocked through inhibition of rho-kinase prior to treatment with latrunculin, blebs no longer emerge (*Charras et al., 2008*; *Figure 6—figure supplements 2 and 3A*). Thus, the growth of latrunculin-induced blebs depends on pressure generated by myosin contractility and they can be used as pressure gauges. When we locally depressurised HeLa cells by establishing a fluidic connection with a micropipette, we found that latrunculin treatment could still induce blebs in both interphase and metaphase cells (*Figure 6E*, *Figure 6—figure supplement 2B*). This effect was independent of the duration over which depressurisation was maintained. When we established a pressure gradient in cells pretreated with Y27632, we did not observe blebs upon latrunculin treatment (*Figure 6F*, *Figure 6—figure supplement 2B*).

As latrunculin triggers blebs indirectly and affects the whole of the actomyosin cytoskeleton, we repeated our experiments using blebs triggered by localised laser ablation of the cortex (*Tinevez et al., 2009*; *Cao et al., 2020*). In control conditions, a short pulse of UV laser focused on the cell cortex of a prometaphase cell led to the emergence of a bleb (*Figure 6G*, *Figure 6—figure supplement 3C*). When myosin contractility was inhibited, ablation did not induce blebs (*Figure 6G*, *Figure 6—figure supplement 3C*), consistent with (*Tinevez et al., 2009*). When a pressure gradient was established, laser ablation could still induce blebs after several minutes (*Figure 6G*, *Figure 6—figure supplement 3C*), indicating that pressure remained sufficiently large for bleb growth. Collectively, these experiments show that intracellular pressure is strongly compartmentalised in cells and that stable pressure gradients can be sustained over durations of several minutes.

## Discussion

Our experiments revealed that cells could sustain pressure gradients across their cytoplasm over durations of 10 min, relevant to cell polarisation and migration. In our experiments, we artificially generated an intracellular pressure gradient with a constant high pressure generated by the cell cortex and

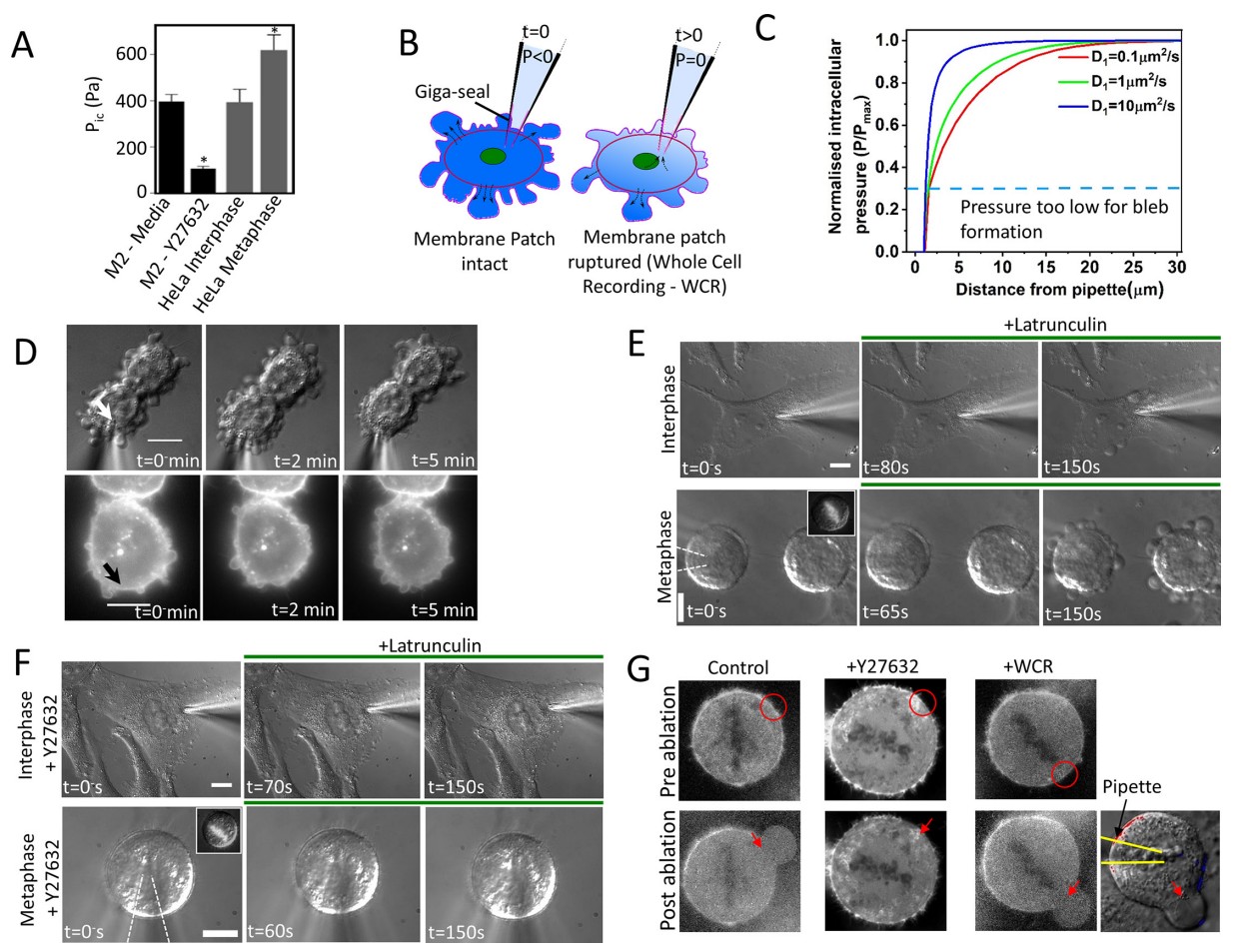

**Figure 6.** Cells can accommodate sustained intracellular pressure gradients. (**A**) Intracellular pressure in Filamin-deficient blebbing M2 melanoma cells and HeLa cells. Cells were treated with Y27632 for 30 min prior to measurement. M2 cells, $n$ = 26 cells per condition from $N$ = 4 experiments. HeLa cells: interphase: $n$ = 65 cells, metaphase: $n$ = 58 cells from $N$ = 4 conditions. Conditions were compared between cells of the same type with a Wilcoxon rank sum test. * indicate significant differences: p = 8 × 10$^{-7}$ for M2 cells, p = 0.001 for HeLa cells. (**B**) Schematic diagram of the pressure release experiment. At time $t$ = 0 s, a fluidic communication is established between a cell and a micropipette, resulting in a small suction pressure at the tip of the pipette. This leads to the establishment of a pressure gradient within the cell. As blebs are pressure-driven protrusions, they can be used as pressure gauges to report on the effect of pressure release. (**C**) Computational prediction of the intracellular pressure profile in response to depressurisation for different membrane-cortex diffusion constants and fixed cytoplasmic diffusion constants. The dashed line represents the pressure below which blebbing cannot occur based on the pressure measured in control M2 cells and cells treated with the Rho-kinase inhibitor Y27632. (**D**) Representative pressure release experiments in M2 cells. Top row: DIC images of a representative experiment. Pressure is released at the pipette tip in a blebbing cell at $t$ = 0 s and maintained constant thereafter. The location of the pipette tip is indicated by a white arrow. A second cell in the field of view serves as a negative control. Scale bar = 10 μm. Bottom row: Fluorescence images of the F-actin cytoskeleton in blebbing cells during a pressure release experiment. The location of the micropipette is indicated by the black arrow. Scale bar = 10 μm. Summary statistics over $n$ = 19 cells are presented in *Figure 6—figure supplement 1*. (**E**) Pressure release experiments in interphase (top) and metaphase (bottom) HeLa cells. Pressure is locally released in the cells at $t$ = 0 s through a micropipette. The presence of intracellular pressure is detected by the emergence of blebs in response to partial depolymerisation of the F-actin cytoskeleton by latrunculin treatment ($t$ = 80 s and $t$ = 150 s). Scale bar = 10 μm. (**F**) Pressure release experiments in interphase (top) and metaphase (bottom) HeLa cells treated with the Rho-kinase inhibitor Y27632. (**E, F**) Drugs were added at $t$ = 0$^{+}$ s. (**G**) Laser ablation of the cortex of metaphase HeLa cells expressing GFP-actin. The target region for laser ablation is indicated by the red circle in the before images and by a red arrow in the after images. Control cells, cells treated with the inhibitor of contractility Y27632 for 30 min, and cells in which a suction was applied through a pipette are shown. (**E–G**) Summary statistics are presented in *Figure 6—figure supplement 3B, C*.

The online version of this article includes the following figure supplement(s) for figure 6:

**Figure supplement 1.** Angular distribution of blebs in a blebbing melanoma cell during a pressure release experiment.

**Figure supplement 2.** Effect of latrunculin treatment on F-actin and myosin distribution within interphase and metaphase HeLa cells.

**Figure supplement 3.** Proportion of cells displaying blebs on their surface in response to Latrunculin treatment.

a low pressure at the tip of a micropipette. Using blebs as reporters of local intracellular pressure, we showed that, despite the presence of an intracellular pressure gradient, pressure over most of the cell periphery remained sufficiently high to continuously generate blebs over several minutes. This result was independent of cell type or cell cycle stage. Computational simulations indicate that the poroelastic properties of the cytoplasm combined with membrane permeability allow the maintenance of stable intracellular pressure gradients.

## Global intracellular flows arise from the combination of cortical tension and a poroelastic cytoplasm

Here, we show that local application of stress to the cell surface induces intracellular water flows spatially distributed over tens of microns and lasting seconds. When we locally deformed the cell surface with an AFM cantilever, we observed a global change in cell height with two different temporal regimes: one quasi-instantaneous and another that equilibrated more slowly, consistent with previous work (*Rosenbluth et al., 2008*). Previous work has shown that the second phase cannot be explained by a linear viscoelastic behaviour (*Rosenbluth et al., 2008*). Therefore, we investigated the role of poroelastic behaviour of the cytoplasm (*Moeendarbary et al., 2013*). In line with this idea, when we decreased the cytoplasmic hydraulic pore size $\xi$, the second phase equilibrated slower taking several seconds. Although we cannot exclude a role for complex mechanotransductory processes, our experiments and simulations suggested that the initial fast relaxation is due to simultaneous motion of the solid and fluid phases over a length-scale controlled by surface tension generated by the cortex and the second slower relaxation arise from relative flows between the two phases equilibrating gradients of fluid pressure (*Moeendarbary et al., 2013*). However, simple scaling arguments for the poroelastic efflux time based on a hydraulic pore size controlled by homogenous deformation of the solid phase in response to volume change underestimated the magnitude of change in $t_p$ ($t_p \sim \frac{1}{D_p} \sim \xi^{-2}$, see Methods). Indeed, these arguments predicted changes of 1.7- to 1.9-fold in $t_p$, much smaller than the ~3.5-fold changes observed in the characteristic equilibration time $\tau_p$ of the slow phase (*Figure 1D*). This is likely because the hydraulic pore size is governed in a complex manner by interplay between multiple solid structures (cytoskeleton, mitochondria, and membrane bounded organelles) and macromolecular crowding.

The existence of spatially distributed intracellular flows induced by local deformation of the cell surface may have important consequences for our understanding of mechanotransduction, which detects mechanical changes and converts them into biochemical signals. Mechanosensory processes act at the molecular scale through opening of ion channels or unfolding of proteins. Thus, the spatial extent of the stress and strain fields will determine what proportion of the cell receives a stimulus sufficiently large and long-lasting to trigger these molecular processes. A challenge for future studies will be to link cellular-scale fluid flows induced by deformation and mechanical stresses to molecular-scale activation of mechanosensory processes to determine the strength of the mechanical stimulus necessary to elicit a whole-cell mechanosensory response. Interestingly, intracellular flows and pressure gradients might participate in triggering signalling through mechanotransductory pathways located away from the cell membrane. For example, Phospholipase A2 activation at the nuclear membrane of epithelial cells is central to the response of tissues to wounding (*Enyedi et al., 2016*). Thus, the intracellular fluid flows revealed by our study might stimulate intracellular mechanosensory mechanisms, representing an indirect mechanotransductory mechanism.

## Cortical contractility, cytoplasmic poroelastic properties, and membrane permeability combine to enable sustained intracellular pressure gradients

Our findings have consequences for our understanding of cell movement in confined environments and low adhesive conditions when cells migrate using blebs (*Bergert et al., 2015*; *Liu et al., 2015*; *Ruprecht et al., 2015*). At steady state, migrating cells form a stable gradient in myosin that increases from front to rear and that powers migration. At the front, protrusion can either consist of a stable bleb when protrusions grow at a rate similar to actin accumulation (*Liu et al., 2015*; *Ruprecht et al., 2015*) or a succession of blebs when actin accumulation rate is faster (*Bergert et al., 2015*; *Bergert et al., 2012*). Disruption of cortical tension at the cell rear by laser ablation leads to cessation of movement as well as localised blebbing in the region of ablation, indicating that myosin accumulation

generates a high tension and high pressure at the rear (*Bergert et al., 2015*). Our work indicates the cell can sustain long-lasting gradients of pressure and raises the possibility that these may participate in migration. In this picture, the observed gradient in actomyosin distribution might generate an intracellular pressure gradient driving a forward-directed intracellular flow, consistent with some experimental observations (*Keren et al., 2009*; *Zicha et al., 2003*). In support of this, recent work has shown that new cell protrusions emerge in regions of the leading edge where membrane–cortex attachment is the weakest, suggesting a role for pressure as a driving force (*Bisaria et al., 2020*). Therefore, pressure gradients may play a direct role in the generation of forward protrusion.

## Conclusions

In summary, our work shows that steady-state gradients in intracellular pressure can arise through the combination of cytoplasmic poroelastic properties and membrane permeability. Thus, intracellular pressure gradients and intracellular fluid flows may play far more important roles than generally appreciated in cell physiology and poroelastic properties must be considered to gain a quantitative understanding of cellular phenomena such as mechanotransduction, cell shape change, and cell migration.

## Materials and methods

### Cell culture

HeLa human cervical cancer cells (HeLa Kyoto) were grown at 37°C with 5% $CO_2$ in DMEM (Life Technologies, UK) supplemented with 10% FBS (Sigma-Aldrich) and 1% penicillin/streptomycin. Melanoma (M2) cells were grown in MEM with Earle's salts (Life Technologies, UK) supplemented with 10% of an 80:20 mix of newborn calf serum: FBS and 1% penicillin/streptomycin. One day before the experiments cells were trypsinised (Trypsin-EDTA, Life Technologies), transferred from tissue culture flasks into glass bottom tissue culture dishes (Willco Wells, The Netherland). Prior to experiments, the medium was replaced with Leibovitz L-15 (Life Technologies, UK) supplemented with 10% FBS.

To visualise the cytoskeleton, the membrane, and the nucleus, we used previously described stable cell lines: HeLa histone mRFP LifeAct GFP, HeLa Life-Act-Ruby, HeLa MRLC-GFP, HeLa CAAX-GFP, and M2 Life-Act-Ruby established using retroviruses and lentiviruses. These were maintained with appropriate selection antibiotics (1 mg/ml G418 and/or 250 ng/ml puromycin).

Cells were routinely tested for the presence of mycoplasma using the mycoALERT kit (Lonza). None of the cell lines in this study were found in the database of commonly misidentified cell lines maintained by ICLAC and NCBI Biosample.

### Metaphase arrest

Cells were cultured to reach 70% confluency before being treated with 10 μM s-trityl-L-cysteine (Sigma-Aldrich) overnight to block them in prometaphase. Cells were washed three times using normal medium and then they were released into 20 μM MG132 (Sigma-Aldrich) for 1–2 hr. After this time, many cells were blocked in metaphase. Before the experiments, the medium was replaced with L-15 containing FBS and the same concentration of MG132.

### Microscopy and laser ablation

Differential interference contrast and epifluorescence imaging was performed on a Nikon TE2000U (Nikon Corp, Japan) inverted microscope. Images were captured on an EMCCD camera (Hamamatsu OrcaER, Hamamatsu, Japan) and transferred to a PC running μmanager (Micromanager, CA). Images were acquired using a 100× oil immersion objective lens (NA = 1.3, Nikon) with 2 × 2 binning. The Ruby fluorophore was imaged using 561 nm excitation and collecting emission at 617 nm. GFP was imaged using 488 nm excitation and collecting emission at 515 nm.

For some experiments, we used an Olympus IX81 inverted microscope equipped with an Olympus FV-1000 scanning laser confocal head. All images were acquired with a 100× oil immersion objective. Imaging of Ruby and mRFP was performed using a 543-nm laser and imaging of GFP was done using a 488-nm laser. The cell membrane was labelled with CellMask Deep red (Thermo Fisher, C10046).

Laser ablation experiments were performed as described in *Tinevez et al., 2009* on a scanning confocal microscope (Olympus FV-1000) equipped with two scanning heads. For ablation, the cortex

of metaphase HeLa cells was exposed to multiple pulses of a 405-nm picosecond pulsed laser (Pico-quant). Following induction, blebs grow rapidly before stopping and eventually retracting.

## Chemical treatments

Drug treatments were carried out by adding the appropriate concentration to the medium. Drugs were present at all times during imaging and patch clamping experiments. Latrunculin B (250 nM Sigma-Aldrich) was used to induce blebbing in the cells by inducing partial loss of F-actin. Y27632 (50 µM, Sigma-Aldrich) was used to inhibit actomyosin contractility. Both drugs were dissolved in DMSO. Vehicle controls were carried out by treating the cells with the same amount of DMSO and for the same duration as in the drug treatment cases.

Sucrose (200 mM, Sigma-Aldrich) was used to increase the medium osmolarity, which resulted in shrinkage of the cells and therefore decreasing the mesh size of cytoplasm. EIPA (50 µM, Sigma-Aldrich), an inhibitor of regulatory volume increase, was used in whole-cell patch-clamp experiments to prevent volume increase due to transportation of solutes into cell.

## Electrophysiology

The experimental equipment setup consisted of a Digidata 1440A Digitizer and a MultiClamp 700B Amplifier piloted with the pCLAMP 10 Software (all from Molecular Devices, CA). Micropipettes were pulled from thin wall borosilicate capillaires (BF100-78-10, Sutter Instruments, CA) using a Flaming/Brown micropipette puller (Model P-97, Sutter Instruments, CA). Micropipettes had resistance of 6.0–6.5 MΩ and a tip diameter of around 2 µm.

For HeLa cells, the pipette was backfilled with a solution was composed of 150 mM K gluconate, 0.005 mM Ca gluconate, 1 mM Mg gluconate, 2 mM K-ATP, 1 mM EGTA, 5 mM HEPES, and 5 mM Glucose with pH = 7.2. For M2 cells, the pipette was backfilled with a solution composed of of 130 mM KCl, 10 mM NaCl, 1 mM $MgCl_2$, 5 mM Na-ATP, 5 mM EGTA, 10 mM HEPES, and 1 mM $CaCl_2$ with pH = 7.2. The bath solution was L15 (Gibco Life Technologies, UK) for both cell types and did not contain FBS because this prevents gigaseal formation.

## Data recording and synchronisation

In patch-clamp experiments, the pressure transmitter, pinch valves, patch-clamp equipment, and the microscope were all connected to the digitiser. This enabled us to control all devices through one platform. The electrophysiology software (pClamp10, Molecular Devices) and the imaging software (Micromanager) were set to communicate to each other. All of the steps to obtain whole-cell configuration were performed manually and a macro was created in pClamp 10 to automatically acquire data once whole-cell configuration was achieved. At this point by starting the macro in pClamp10, data acquisition, the timing of acquisition of each image, the timing of opening and closing of pinch valves as well as pressure measurement were recorded automatically through one software. This enabled us to synchronise all devices and determine the exact image at which pressure was applied to the micro-pipette and injection started.

In AFM experiments, for imaging, the camera (Hamamatsu OrcaER, Hamamatsu, Japan) was triggered every 100 ms using one of the output channels of the digitiser. The cantilever displacement and force were also acquired by connecting the corresponding output channels of the AFM to the digitiser using BNC cables. Therefore, similar to patch-clamp experiments, we were able to synchronise all equipment by integrating them into one acquisition platform.

## Measurement of intracellular pressure

Direct measurements of intracellular pressure were effected using the 900A micropressure system (World Precision Instruments) according to the manufacturer's instructions and as described in *Petrie et al., 2014*. Briefly, a 0.5-µm micropipette (World Precision Instruments) was filled with a 1 M KCl solution, placed in a microelectrode holder half-cell (World Precision Instruments), and connected to a pressure source regulated by the 900A system. A calibration chamber (World Precision Instruments) was filled with 0.1 M KCl and connected to the 900A system, and the resistance of each microelectrode was set to zero and then secured in a MPC-325 micromanipulator (Sutter Instrument) within an environmental chamber (37°C and 10% $CO_2$) on an Axiovert 200M microscope (ZEISS). To measure intracellular pressure, the microelectrode was driven at a 45° angle into the cytoplasm, maintained in

place for ≥5 s before being removed. The pressure measurement was calculated as the mean pressure reading during this interval of time.

## Microinjection and pressure release setup

A pressure sensor (IMPRESS sensors and systems, IMP-LR-C0238-7A4-BAV-00-000) was used to measure the magnitude and temporal evolution of applied pressure. A glass Erlenmeyer was used as a pressure reservoir and was connected to a plastic tube which could be opened by a computer-controlled pinch valve. The tube was then connected to the pressure sensor and the micropipette holder. Two digital manometers were used to monitor the pressure in the reservoir and just before the pipette holder. The length of all tubing was approximately 1 m and the time for pressure to propagate through the tubes, plus the response time of switches was around 30 ms (data not shown), which is less than our frame interval (66.7 or 100 ms). The pressure transmitter, pinch valves, patch-clamp equipment, and the microscope were all connected to the digitiser. Before the start of the experiments, pressure in the reservoir was set. Once the whole-cell patch clamping configuration was achieved, a pulse of pressure was applied by opening the valve, resulting in injection of fluid into the cell. In pressure release experiments, after forming a whole-cell configuration, opening the valve resulted in connecting the pipette to the open atmosphere with pressure of zero ($P_{atm} = 0$).

## Statistical analysis

All statistical analysis was carried using Microsoft Excel. In all graphs, error bars indicate standard deviation. In box plots, the whiskers represent range of data.

## Image processing

Fiji was used for producing kymographs and preparation of images for the figures.

To extract the profile of the cell surface for *Figure 4—figure supplement 1*, we cropped the top half of the cell and, for each *x* position, we determined the *z* position of the maximum fluorescence intensity closest to the cell interior. Then, to determine the profile of indentation, we subtracted the position during indentation from the one before indentation at each *x* position.

## Functionalisation of fluorescent beads

Yellow-green carboxylate-modified fluorescent nanobeads with a diameter of 500 nm (FluoSpheres, Molecular Probes, Invitrogen) were coated with collagen-I following the manufacturer's protocol. To attach nanobeads to the cell membrane prior to experiments, HeLa cells were incubated for ~30 min with a dilute solution (1:100 dilution) containing the fluorescent collagen-coated nanobeads. Unattached beads were then washed out prior to experimentation.

## Defocusing microscopy

Collagen-coated fluorescent beads were added to cell culture dishes 30 min before the start of the experiments. Some of the beads attached to the cells. The cells were washed in L15 three times to remove unattached beads. Imaging solution was Leibovitz L-15 (Life Technologies, UK) supplemented with 10% FBS. Defocusing microscopy was implemented as described in *Rosenbluth et al., 2008*. A cell with two or three attached beads was selected. The motion of integrin-bound fluorescent beads was tracked in three dimensions using defocusing fluorescence microscopy. By focusing a few microns above the bead plane, each bead appeared as a set of concentric rings. The distance between the beads and image plane is directly related to the radius of the outer ring and is used to determine relative *z* displacements. Time-lapse images were acquired every 67 or 100 ms and stored as a stack for each bead.

A code was written in MATLAB to automatically track the motion of the beads. Briefly, for a selected bead, a line intensity profile along the diameter of the concentric circles was obtained for each image in the stack. The radius of the outer ring was obtained from the Gaussian fit to the first and last peak in each image. For calibration, changes in radius were then converted into changes Z by moving a bead by a known distance using a piezoelectric stage (*Esteki et al., 2021*) and acquiring images.

## Atomic force microscopy and data analysis

Indentations of cells by AFM were performed using a JPK NanoWizard-1 AFM (JPK, Berlin, Germany) mounted on an inverted microscope (IX-81, Olympus, Berlin, Germany). The day prior

to experimentation, cells were plated onto 35 mm glass bottom Petri dishes. Experiments were performed at room temperature and cells were maintained in Leibovitz L15 medium (Life Technologies) supplemented with 10% FBS (Sigma-Aldrich) and MG132 (10 μM). Before each experiment, the spring constant of the cantilever was calibrated using the thermal noise method implemented in the AFM software (JPK SPM). The sensitivity of the cantilever was measured from the slope of force–distance curves acquired on glass. For apparent stiffness measurements, we used soft cantilevers with V-shaped tips (BioLever OBL-10, Bruker; nominal spring constant of 0.006 N m⁻¹).

For each measurement, the cantilever was first aligned above the cell of interest using the optical microscope. Then, it was lowered towards the cell with an approach speed of 10 μm/s until reaching a force setpoint of 5 nN and then kept the cantilever at a constant height.

## FE modelling

We conducted FE simulations to model the mechanical response at the cell scale, specifically focusing on the local deformations of the living cell surface and pressure gradients driving intracellular cytosolic flows. These simulations aimed to capture the deformation behaviour of a poroelastic cell when subjected to changes in effective pressure, mimicking scenarios such as fluid injection or pressure release in one region of the cell surface. FE models were developed using ABAQUS (version 2018). We used nonlinear geometry and unstructured mesh in our FE simulations, and also took into consideration a neo-Hookean isotropic porohyperelastic model (*Esteki et al., 2021*). This was due to the large mechanical deformation range observed during our AFM, microinjection, and pressure release experiments. The best mesh and domain sizes were determined via mesh convergence studies, and a tolerance for the maximum pore pressure change per increment was calculated for SOILS analysis in our simulations.

### AFM microindentation simulations

We ran simulations on two-dimensional rectangular sections to computationally investigate the influence of poroelasticity in the temporal mechanical response of cells to indentation (*Figure 4A*). To minimise edge effects, the cell was represented as a cylindrical disk (20 μm radius, 20 μm thickness) indented by 2 μm with an infinitely rigid indenter representing the AFM cantilever tip. Frictionless and impermeable contact between the indenter and the cell was assumed, and a no-slip condition was imposed on the bottom surface of the cylinder. Pore pressure was set to zero except at the indenter contact surface to simulate fluid drainage. The simulation domain was discretised using quadratic quadrilateral CAX8P elements. Mesh sensitivity checks were performed to ensure independence of results on element size. The simulation consisted of a ramp step followed by a hold step. The ramp step entails indenting the material surface instantaneously (~0.01 s) with specific indentation displacement (~2 μm). The phase in which the force reaction is released is known as the hold step. The main parameter of interest was displacement of the cell surface in response to localised indentation. The force–relaxation curve and final displacement of the cell surface as a function of distance from the AFM tip were analysed to extract the elastic modulus and poroelastic diffusion coefficient by fitting to the experimental data (for a thorough approach and curve fitting techniques, see *Esteki et al., 2020*).

### Fluid injection and pressure release

We ran FE simulations of a poroelastic cell that responds to the application of effective pressure changes at its top boundary where the micropipette contact its surface. The initial cell shape was idealised based on confocal profiles (*Figure 1B*). We modelled the cell as an axisymmetric elliptical cap with a diameter of 40 μm and a thickness of 4.5 μm attached to a substrate. The pipette was in contact with the top surface in the centre of the cell sufficiently long to enable equilibration of effective pressure with the same value as the endogenous cell pressure, $P_{eff} = P_{app} - P_{in} = 0$. The difference between internal $(P_{in})$ and applied $(P_{app})$ pressure induces pressure gradients $(P_{eff})$ across the pipette–cell boundary and causes fluid to flow across this boundary. For fluid injection, $P_{eff} > 0 \rightarrow P_{app} > P_{in}$, and for fluid releasing $P_{eff} < 0 \rightarrow P_{app} < P_{in}$ were considered. *Figure 4B* shows that the pressure increase within the pipette at $t = 0$ s led to fluid injection into the cell causing a rise in surface height and pressure changes until it reaches steady state, $P_{eff} = P_{app} - P_{in} = 0$. Our simulation had a ramp time ($t = 2$ s) over which applied pressure increased the internal pressure. The poroelastic material domain was discretised using the quadratic quadrilateral CPE8P element

and the sensitivity of the FE simulations to domain size and mesh element numbers were checked. When the injection is applied, the cell surface reacts with a time lag ($\delta_t$) that increases with the the distance from the micropipette. $\delta_t$ is defined as the time for which surface movement reaches 10% of the steady-state displacement. In the FE simulations, the elastic modulus—which was extracted from AFM microindentation tests—was taken into account to fit the experimental fluid injection curves and derive $P_{eff}$ and $D$. To show the impact of the membrane–cortex layer on intracellular pressure gradients and pressure compartmentalisation in living cells, we modelled cells as a double layer poroelastic material (*Charras et al., 2009*). The cytoplasmic material is parameterised by its elasticity $E_2$, poroelastic diffusion constant $D_2$, and pressure $P_{in}$ and it is surrounded by a less permeable thin layer representing the cortex parameterised by $E_1$, $D_1$, and $P_{in}$. A no-slip condition was imposed between the two layers.

## Parameterisation of the porohyperelastic model

As in previous studies, we used a fixed Poisson ratio of $\upsilon = 0.3$ (*Esteki et al., 2021*) and extracted elastic modulus and diffusion constant by fitting experimental steady-state vertical displacement of beads tethered to the cell surface in response to localised indentation with FE simulation (*Figure 4C*). Considering the surface displacement curve, we extracted the following elastic modulus and diffusion constant, $E = 1.8$ kPa and $D = 28$ µm$^2$ s$^{-1}$, consistent with our experimental results and previous work. To determine effective pressure ($P_{eff}$), elastic modulus ($E$), and poroelastic diffusion constant ($D$) for the fluid microinjection of the HeLa cells, we used optimisation processes to fit the vertical displacement of nanobeads positioned on the cell surface after 2 s microinjection with FE simulations (*Figure 4D*). The first step in the optimisation process was to conduct the simulations to match the experimental curves and get a first approximations of $D$ and $E$ taking into account $P_{eff} = 500$ Pa based on experimental measurements. Next, we adjusted $E$ and $D$ to minimise error compared to the experimental curves and this yielded $D = 13$ µm$^2$ s$^{-1}$ and $E = 1.2$ kPa.

## Scaling of the poroelastic relaxation time with pore size

To predict changes in poroelastic relaxation time with cell volume, we tried to gain insight using simple scaling arguments. The poroelastic diffusion constant scales as $D_p \sim \xi^2$, with $\xi$ the hydraulic pore size, and the poroelastic fluid efflux time $t_p$ scales as $t_p \sim \frac{1}{D_p} \sim \xi^{-2}$. Previous work showed that HeLa cell volume decreases by ~40% in response to hyperosmotic shock (*Charras et al., 2009*). The fluid volume fraction $V_f$ in cells is ~65–75%. If we assume that intracellular water is contained in $N$ pores of volume $\xi_0^3$, we can express the cell volume as $V_0 = V_S + N.\xi_0^3$ with $V_s$ the volume of the solid fraction. We can rewrite $V_S \sim \phi.V_f \sim \phi.N.\xi_0^3$ with $\phi = [0.42, 0.6]$. As $V_s$ does not change in response to osmotic shock, we can rewrite the volume change $\frac{V}{V_0} = \alpha$ to obtain the change in pore size $\frac{\xi}{\xi_0} = (\alpha + \phi(\alpha - 1))^{\frac{1}{3}} = [0.72, 0.77]$ for $\alpha = 0.4$. This leads to an estimated change in $\frac{t_p}{t_{p0}} \sim [1.7, 1.9]$.

## Acknowledgements

The authors are grateful to members of the GC lab for feedback and suggestions on the manuscript. The authors wish to acknowledge Prof Guillaume Salbreux and Dr Pragya Srivastava for insightful discussions. MM and GC were supported by grant WT092825 from the Wellcome Trust. GC was supported by a University Research Fellowship from the Royal Society.

## Additional information

### Funding

| Funder | Grant reference number | Author |
| --- | --- | --- |
| Royal Society | URF | Guillaume Charras |
| Wellcome Trust | 10.35802/092825 | Guillaume Charras |

| Funder | Grant reference number | Author |
|---|---|---|

The funders had no role in study design, data collection, and interpretation, or the decision to submit the work for publication. For the purpose of Open Access, the authors have applied a CC BY public copyright license to any Author Accepted Manuscript version arising from this submission.

## Author contributions

Majid Malboubi, Conceptualization, Data curation, Formal analysis, Investigation, Methodology, Writing – original draft, Writing – review and editing; Mohammad Hadi Esteki, Formal analysis, Investigation, Methodology, Writing – review and editing; Malti B Vaghela, Conceptualization, Formal analysis, Investigation, Methodology; Lulu IT Korsak, Data curation, Formal analysis, Methodology; Ryan J Petrie, Data curation, Formal analysis, Methodology, Writing – review and editing; Emad Moeendarbary, Conceptualization, Funding acquisition, Methodology, Writing – review and editing; Guillaume Charras, Conceptualization, Funding acquisition, Methodology, Writing – original draft, Project administration, Writing – review and editing

## Author ORCIDs

Guillaume Charras (iD) https://orcid.org/0000-0002-7902-0279

Reviewer #1 (Public review): https://doi.org/10.7554/eLife.105523.3.sa1
Reviewer #2 (Public review): https://doi.org/10.7554/eLife.105523.3.sa2
Reviewer #3 (Public review): https://doi.org/10.7554/eLife.105523.3.sa3
Author response https://doi.org/10.7554/eLife.105523.3.sa4

---

# Additional files

## Supplementary files
MDAR checklist

## Data availability
The cell lines used in this study are available from the corresponding author upon request. Images, analysis scripts, and image quantification data are deposited in the UCL research data repository (https://rdr.ucl.ac.uk) and associated with the DOI https://doi.org/10.5522/04/26770204.

The following dataset was generated:

| Author(s) | Year | Dataset title | Dataset URL | Database and Identifier |
|---|---|---|---|---|
| Malboubi M, Esteki MH, Vaghela M, Korsak L, Petrie R, Moeendarbary E, Charras G | 2025 | The cytoplasm of living cells can sustain transient and steady intracellular pressure gradients | https://doi.org/10.5522/04/26770204 | UCL Research Data Repository, 10.5522/04/26770204 |

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
