## [Editor Report · eLife Assessment]

This **important** study combines imaginative and innovative experiments with a finite element modelling to demonstrate the relevance of poroelasticity in the mechanical properties of cells across physiologically relevant time and length scales. The authors present **convincing** evidence that cytosolic flows and pressure gradients can persist in cells with permeable membranes, generating spatially segregated influx and outflux zones. These findings are of interest to the cell biology and biophysics communities.

---

## [Referee Report · Reviewer #1 (Public review)]

Summary:

This work investigated whether cytoplasmic poroelastic properties play an important role in cellular mechanical response over length scales and time scales relevant to cell physiology. Overall, the manuscript concludes that intracellular cytosolic flows and pressure gradients are important for cell physiology and that they act of time- and length-scales relevant to mechanotransduction and cell migration.

Strengths:

Their approach integrates both computational and experimental methods. The AFM deformation experiments combined with measuring z-position of beads is a challenging yet compelling method to determine poroelastic contributions to mechanical realization.

The work is quite interesting and will be of high value to the field of cell mechanics and mechanotransduction.

Weaknesses:

The weaknesses I noted earlier were adequately addressed in the revised version.

---

## [Referee Report · Reviewer #2 (Public review)]

Summary:

Malboubi et al. present an experimental framework to investigate the rheological properties of the cell cytoplasm. Their findings support a model where the cytoplasm behaves as a poroelastic material governed by Darcy's law. They demonstrate that this poroelastic behavior delays the equilibration of hydrostatic pressure gradients within the cytoplasm over timescales of 1 to 10 seconds following a perturbation, likely due to fluid-solid friction within the cytoplasmic matrix. Furthermore, under sustained perturbations such as depressurization, they reveal that pressure gradients can persist for minutes, which they propose might potentially influence physiological processes like mechanotransduction or cell migration typically happening on these timescales.

Strengths:

This article holds significant value within the ongoing efforts of the cell biology and biophysics communities to quantitatively characterize the mechanical properties of cells. The experiments are innovative and thoughtfully contextualized with quantitative estimates and a finite element model that supports the authors' hypotheses.

Comments & Questions:

The authors have successfully addressed the questions and comments raised in my previous review, significantly improving the manuscript's depth. Regarding my last question on the predicted saturation of the time lag, the authors propose the interesting hypothesis that the cell cortex becomes dominant at distances beyond 30 microns and plan to test this hypothesis at a later stage.

---

## [Referee Report · Reviewer #3 (Public review)]

Summary:

In this delightful study, the authors use local indentation of the cell surface combined with out-of-focus microscopy to measure the rates of pressure spread in the cell and to argue that the results can be explained with the poroelastic model. Osmotic shock that decreases cytoskeletal mesh size supports this notion. Experiments with water injection and water suction further support it, and also, together with a mechanical model and elegant measurements of decreasing fluorescence in the cell 'flashed' by external flow, demonstrate that the membrane is permeable, and that steady flow and pressure gradient can exist in a cell with water source/sink in different locations. Use of blebs as indicators of the internal pressure further supports the notion of differential cytoplasmic pressure.

Strengths:

The study is very imaginative, interesting, novel and important.

Weaknesses: I have two broad critical comments:

(1) I sense that the authors are correct that the best explanation of their results is the passive poroelastic model. Yet, to be thorough, they have to try to explain the experiments with other models and show why their explanation is parsimonious. For example, one potential explanation could be some mechanosensitive mechanism that does not involve cytoplasmic flow; another could be viscoelastic cytoskeletal mesh, again not involving poroelasticity. I can imagine more possibilities. Basically, be more thorough in the critical evaluation of your results. Besides, discuss potential effect of significant heterogeneity of the cell.

(2) The study is rich in biophysics but a bit light on chemical/genetic perturbations. It could be good to use low levels of chemical inhibitors for, for example, Arp2/3, PI3K, myosin etc, and see the effect and try to interpret it. Another interesting question - how adhesive strength affects the results. A different interesting avenue - one can perturb aquaporins. Etc. At least one perturbation experiment would be good.

Comments on revisions: I am satisfied with the revisions

---

## [Author Response]

The following is the authors’ response to the original reviews

**Reviewer #1 (Public review):**
(1) Some details are not described for experimental procedures. For example, what were the pharmacological drugs dissolved in, and what vehicle control was used in experiments? How long were pharmacological drugs added to cells?

We apologise for the oversight. These details have now been added to the methods section of the manuscript as well as to the relevant figure legends.

Briefly, latrunculin was used at a final concentration of 250 nM and Y27632 at a final concentration of 50 μM. Both drugs were dissolved in DMSO. The vehicle controls were effected with the highest final concentration of DMSO of the two drugs.

The details of the drug treatments and their duration was added to the methods and to figures 6, S10, and S12.

(2) Details are missing from the Methods section and Figure captions about the number of biological and technical replicates performed for experiments. Figure 1C states the data are from 12 beads on 7 cells. Are those same 12 beads used in Figure 2C? If so, that information is missing from the Figure 2C caption. Similarly, this information should be provided in every figure caption so the reader can assess the rigor of the experiments. Furthermore, how heterogenous would the bead displacements be across different cells? The low number of beads and cells assessed makes this information difficult to determine.

We apologise for the oversight. We have now added this data to the relevant figure panels.

To gain a further understanding of the heterogeneity of bead displacements across cells, we have replotted the relevant graphs using different colours to indicate different cells. This reveals that different cells appear to behave similarly and that the behaviour appears controlled by distance to the indentation or the pipette tip rather than cell identity.

We agree with the reviewer that the number of cells examined is low. This is due to the challenging nature of the experiments that signifies that many attempts are necessary to obtain a successful measurement.

The experiments in Fig 1C are a verification of a behaviour documented in a previous publication [1]. Here, we just confirm the same behaviour and therefore we decided that only a small number of cells was needed.

The experiments in Fig 2C (that allow for a direct estimation of the cytoplasm’s hydraulic permeability) require formation of a tight seal between the glass micropipette and the cell, something known as a gigaseal in electrophysiology. The success rate of this first step is 10-30% of attempts for an experienced experimenter. The second step is forming a whole cell configuration, in which a hydraulic link is formed between the cell and the micropipette. This step has a success rate of ~ 50%. Whole cell links are very sensitive to any disturbance. After reaching the whole cell configuration, we applied relatively high pressures that occasionally resulted in loss of link between the cell and the micropipette. In summary, for the 12 successful measurements, hundreds of unsuccessful attempts were carried out.

(3) The full equation for displacement vs. time for a poroelastic material is not provided. Scaling laws are shown, but the full equation derived from the stress response of an elastic solid and viscous fluid is not shown or described.

We thank the reviewer for this comment. Based on our experiments, we found that the cytoplasm behaves as a poroelastic material. However, to understand the displacements of the cell surface in response to localised indentation, we show that we also need to take the tension of the submembranous cortex into account. In summary, the interplay between cell surface tension generated by the cortex and the poroelastic cytoplasm controls the cell behaviour. To our knowledge, no simple analytical solutions to this type problem exist.

In Fig 1, we show that the response of the cell to local indentation is biphasic with a short time-scale displacement followed by a longer time-scale one. In Figs 2 and 3, we directly characterise the kinetics of cell surface displacement in response to microinjection of fluid. These kinetics are consistent with the long time-scale displacement but not the short time-scale one. Scaling considerations led us to propose that tension in the cortex may play a role in mediating the short time-scale displacement. To verify this hypothesis, we have now added new data showing that the length-scale of an indentation created by an AFM probe depends on tension in the cortex (Fig S5).

In a previous publication [2], we derived the temporal dynamics of cell surface displacement for a homogenous poroelastic material in response to a change in osmolarity. In the current manuscript, the composite nature of the cell (membrane, cortex, cytoplasm) needs to be taken into account as well as a realistic cell shape. Therefore, we did not attempt to provide an analytical solution for the displacement of the cell surface versus time in the current work. Instead, we turned to finite element modelling to show that our observations are qualitatively consistent with a cell that comprises a tensed submembranous actin cortex and a poroelastic cytoplasm (Fig 4). We have now added text to make this clearer for the reader.

**Reviewer #2 (Public review):**
Comments & Questions:The authors state, "Next, we sought to quantitatively understand how the global cellular response to local indentation might arise from cellular poroelasticity." However, the evidence presented in the following paragraph appears more qualitative than strictly quantitative. For instance, the length scale estimate of ~7 μm is only qualitatively consistent with the observed ~10 μm, and the timescale 𝜏𝑧 ≈ 500 ms is similarly described as "qualitatively consistent" with experimental observations. Strengthening this point would benefit from more direct evidence linking the short timescale to cell surface tension. Have you tried perturbing surface tension and examining its impact on this short-timescale relaxation by modulating acto-myosin contractility with Y-27632, depolymerizing actin with Latrunculin, or applying hypo/hyperosmotic shocks?

Upon rereading our manuscript, we agree with the reviewer that some of our statements are too strong. We have now moderated these and clarified the goal of that section of the text.

The reviewer asks if we have examined the effect of various perturbations on the short time-scale displacements. In our experimental conditions, we cannot precisely measure the time-scale of the fast relaxation because its duration is comparable to the frame rate of our image acquisition. However, we examined the amplitude of the displacement of the first phase in response to sucrose treatment and we have carried out new experiments in which we treat cells with 250nM Latrunculin to partially depolymerise cellular F-actin. Neither of these treatments had an impact on the amplitude of vertical displacements (Fig. S3).

The absence of change in response to Latrunculin may be because the treatment decreases both the elasticity of the cytoplasm and the cortical tension . As the length-scale of the deformation of the surface scales as \begin{document}$l \sim \frac{\gamma}{E}$\end{document}, the two effects of latrunculin treatment may therefore compensate one another and result in only small changes in . We have now added this data to supplementary information and comment on this in the text.

The reviewer’s comment also made us want to determine how cortical tension affects the length-scale of the cell surface deformation created by localised microindentation. To isolate the role of the cortex from that of cell shape, we decided to examine rounded mitotic cells. In our experiments, we indented a mitotic cell expressing a membrane targeted GFP with a sharp AFM tip (Fig. S5).

In our experiments, we adjusted force to generate a 2μm depth indentation and we imaged the cell profile with confocal microscopy before and during indentation. Segmentation of this data allowed us to determine the cell surface displacement resulting from indentation and measure a length scale of deformation. In control conditions, the length scale created by deformation is on the order of 1.2μm. When we inhibited myosin contractility with blebbistatin, the length-scale of deformation decreased significantly to 0.8 μm, as expected if we decrease the surface tension γ without affecting the cytoplasmic elasticity. We have now added this data to our manuscript.

The authors demonstrate that the second relaxation timescale increases (Figure 1, Panel D) following a hyperosmotic shock, consistent with cytoplasmic matrix shrinkage, increased friction, and consequently a longer relaxation timescale. While this result aligns with expectations, is a seven-fold increase in the relaxation timescale realistic based on quantitative estimates given the extent of volume loss?

We thank the reviewer for this interesting question. Upon re-examining our data, we realised that the numerical values in the text related to the average rather than the median of our measurements. The median of the poroelastic time constant increases from ~0.4s in control conditions to 1.4s in sucrose, representing approximately a 3.5 fold increase.

Previous work showed that HeLa cell volume decreases by ~40% in response to hyperosmotic shock [3]. The fluid volume fraction \begin{document}$V_{f}$\end{document} in cells is ~65-75%. If we assume that the water is contained in N pores of volume \begin{document}$\xi_{0}^{3}$\end{document}, we can express the cell volume as \begin{document}$V_{0}=V_{S}+N . \xi_{0}^{3}$\end{document} with \begin{document}$V_{S}$\end{document} the volume of the solid fraction. We can rewrite \begin{document}$V_{S} \sim \phi . V_{f} \sim \phi . N . \xi_{0}^{3}$\end{document}.

With ∅ = 0.42 -0.6. As does not change in response to osmotic shock, we can rewrite the volume change \begin{document}$\frac{v}{v_{0}}=\alpha$\end{document} to obtain the change in pore size \begin{document}$\frac{\xi}{\xi_{0}}=(\alpha+\phi(\alpha-1))^{1 / 3=[0.72,0.77]}$\end{document}.

The poroelastic diffusion constant scales as \begin{document}$D_{p} \sim \xi^{2}$\end{document} and the poroelastic timescale scales as \begin{document}$t_{p} \sim \frac{1}{D_{p}} \sim \xi^{-2}$\end{document}. Therefore, the measured change in volume leads to a predicted increase in poroelastic diffusion time of 1.7-1.9 fold, smaller than observed in our experiments. This suggests that some intuition can be gained in a straightforward manner assuming that the cytoplasm is a homogenous porous material.

However, the reality is more complex and the hydraulic pore size is distinct from the entanglement length of the cytoskeleton mesh, as we discussed in a previous publication [4]. When the fluid fraction becomes sufficiently small, macromolecular crowding will impact diffusion further and non-linearities will arise. We have now added some of these considerations to the discussion.

If the authors' hypothesis is correct, an essential physiological parameter for the cytoplasm could be the permeability k and how it is modulated by perturbations, such as volume loss or gain. Have you explored whether the data supports the expected square dependency of permeability on hydraulic pore size, as predicted by simple homogeneity assumptions?

We thank the reviewer for this comment. As discussed above, we have explored such considerations in a previous publication (see discussion in [4]). Briefly, we find that the entanglement length of the F-actin cytoskeleton does play a role in controlling the hydraulic pore size but is distinct from it. Membrane bounded organelles could also contribute to setting the pore size. In our previous publication, we derived a scaling relationship that indicates that four different length-scales contribute to setting cellular rheology: the average filament bundle length, the size distribution of particles in the cytosol, the entanglement length of the cytoskeleton, and the hydraulic pore size. Many of these length-scales can be dynamically controlled by the cell, which gives rise to complex rheology. We have now added these considerations to our discussion.

Additionally, do you think that the observed decrease in k in mitotic cells compared to interphase cells is significant? I would have expected the opposite naively as mitotic cells tend to swell by 10-20 percent due to the mitotic overshoot at mitotic entry (see Son Journal of Cell Biology 2015 or Zlotek Journal of Cell Biology 2015).

We thank the reviewer for this interesting question. Based on the same scaling arguments as above, we would expect that a 10-20% increase in cell volume would give rise to 10-20% increase in diffusion constant. However, we also note that metaphase leads to a dramatic reorganisation of the cell interior and in particular membrane-bounded organelles. In summary, we do not know why such a decrease could take place. We now highlight this as an interesting question for further research.

Based on your results, can you estimate the pore size of the poroelastic cytoplasmic matrix? Is this estimate realistic? I wonder whether this pore size might define a threshold above which the diffusion of freely diffusing species is significantly reduced. Is your estimate consistent with nanobead diffusion experiments reported in the literature? Do you have any insights into the polymer structures that define this pore size? For example, have you investigated whether depolymerizing actin or other cytoskeletal components significantly alters the relaxation timescale?

We thank the reviewer for this comment. We cannot directly estimate the hydraulic pore size from the measurements performed in the manuscript. Indeed, while we understand the general scaling laws, the prefactors of such relationships are unknown.

We carried out experiments aiming at estimating the hydraulic pore size in previous publications [3,4] and others have shown spatial heterogeneity of the cytoplasmic pore size [5]. In our previous experiments, we examined the diffusion of PEGylated quantum dots (14nm in hydrodynamic radius). In isosmotic conditions, these diffused freely through the cell but when the cell volume was decreased by a hyperosmotic shock, they no longer moved [3,4]. This gave an estimate of the pore radius of ~15nm.

Previous work has suggested that F-actin plays a role in dictating this pore size but microtubules and intermediate filaments do not [4].

There are no quantifications in Figure 6, nor is there a direct comparison with the model. Based on your model, would you expect the velocity of bleb growth to vary depending on the distance of the bleb from the pipette due to the local depressurization? Specifically, do blebs closer to the pipette grow more slowly?

We apologise for the oversight. The quantifications are presented in Fig S10 and Fig S12. We have now modified the figure legends accordingly.

Blebs are very heterogenous in size and growth velocity within a cell and across cells in the population in normal conditions [6]. Other work has shown that bleb size is controlled by a competition between pressure driving growth and actin polymerisation arresting it[7]. Therefore, we did not attempt to determine the impact of depressurisation on bleb growth velocity or size.

In experiments in which we suddenly increased pressure in blebbing cells, we did notice a change in the rate of growth of blebs that occurred after we increased pressure (Author response image 1). However, the experiments are technically challenging and we decided not to perform more.

**Author response image 1. sa4fig1:** A. A hydraulic link is established between a blebbing cell and a pipette. At time t>0, a step increase in pressure is applied. B. Kymograph of bleb growth in a control cell (top) an in a cell subjected to a pressure increase at t=0s (bottom). Top: In control blebs, the rate of growth is slow and approximately constant over time. The black arrow shows the start of blebbing. Bottom: The black arrow shows the start of blebbing. The dashed line shows the timing of pressure application and the red arrow shows the increase in growth rate of the bleb when the pressure increase reaches the bleb. This occurs with a delay δt.

I find it interesting that during depressurization of the interphase cells, there is no observed volume change, whereas in pressurization of metaphase cells, there is a volume increase. I assume this might be a matter of timescale, as the microinjection experiments occur on short timescales, not allowing sufficient time for water to escape the cell. Do you observe the radius of the metaphase cells decreasing later on? This relaxation could potentially be used to characterize the permeability of the cell surface.

We thank the reviewer for this comment.

First, we would like to clarify that both metaphase and interphase cells increase their volume in response to microinjection. The effect is easier to quantify in metaphase cells because we assume spherical symmetry and just monitor the evolution of the radius (Fig 3). However, the displacement of the beads in interphase cells (Fig 2) clearly shows that the cell volume increases in response to microinjection. For both interphase and metaphase cells, when the injection is prolonged, the membrane eventually detaches from the cortex and large blebs form until cell lysis. In contrast to the reviewer’s intuition, we never observe a relaxation in cell volume, probably because we inject fluid faster than the cell can compensate volume change through regulatory mechanisms involving ion channels.

When we depressurise metaphase cells, we do not observe any change in volume (Fig S10). This contrasts with the increase that we observe upon pressurisation. The main difference between these two experiments is the pressure differential. During depressurisation experiments, this is the hydraulic pressure within the cell ~500Pa (Fig 6A); whereas during pressurisation experiments, this is the pressure in the micropipette, ranging from 1.4-10 kPa (Fig 3). We note in particular that, when we used the lowest pressures in our experiments, the increase in volume was very slow (see Fig 3C). Therefore, we agree with the reviewer that it is likely the magnitude of the pressure differential that explains these differences.

I am curious about the saturation of the time lag at 30 microns from the pipette in Figure 4, Panel E for the model's prediction. A saturation which is not clearly observed in the experimental data. Could you comment on the origin of this saturation and the observed discrepancy with the experiments (Figure E panel 2)? Naively, I would have expected the time lag to scale quadratically with the distance from the pipette, as predicted by a poroelastic model and the diffusion of displacement. It seems weird to me that the beads start to move together at some distance from the pipette or else I would expect that they just stop moving. What model parameters influence this saturation? Does membrane permeability contribute to this saturation?

We thank the reviewer for pointing this out. In our opinion, the saturation occurring at 30 microns arises from the geometry of the model. At the largest distance away from the micropipette, the cortex becomes dominant in the mechanical response of the cell because it represents an increasing proportion of the cellular material.

To test this hypothesis, we will rerun our finite element models with a range of cell sizes. This will be added to the manuscript at a later date.

**Reviewer #3 (Public review):**
Weaknesses: I have two broad critical comments:(1) I sense that the authors are correct that the best explanation of their results is the passive poroelastic model. Yet, to be thorough, they have to try to explain the experiments with other models and show why their explanation is parsimonious. For example, one potential explanation could be some mechanosensitive mechanism that does not involve cytoplasmic flow; another could be viscoelastic cytoskeletal mesh, again not involving poroelasticity. I can imagine more possibilities. Basically, be more thorough in the critical evaluation of your results. Besides, discuss the potential effect of significant heterogeneity of the cell.

We thank the reviewer for these comments and we agree with their general premise.

Some observations could qualitatively be explained in other ways. For example, if we considered the cell as a viscoelastic material, we could define a time constant \begin{document}$\tau_{R}=\frac{\eta}{E}$\end{document} with η the viscosity and E the elasticity of the material. The increase in relaxation time with sucrose treatment could then be explained by an increase in viscosity. However, work by others has previously shown that, in the exact same conditions as our experiment, viscoelasticity cannot account for the observations[1]. In its discussion, this study proposed poroelasticity as an alternative mechanism but did not investigate that possibility. This was consistent with our work that showed that the cytoplasm behaves as a poroelastic material and not as a viscoelastic material [4]. Therefore, we decided not to consider viscoelasticity as possibility. We now explain this reasoning better and have added a sentence about a potential role for mechanotransductory processes in the discussion.

(2) The study is rich in biophysics but a bit light on chemical/genetic perturbations. It could be good to use low levels of chemical inhibitors for, for example, Arp2/3, PI3K, myosin etc, and see the effect and try to interpret it. Another interesting question - how adhesive strength affects the results. A different interesting avenue - one can perturb aquaporins. Etc. At least one perturbation experiment would be good.

We agree with the reviewer. In our previous studies, we already examined what biological structures affect the poroelastic properties of cells [2,4]. Therefore, the most interesting aspect to examine in our current work would be perturbations to the phenomenon described in Fig 6G and, in particular, to investigate what volume regulation mechanisms enable sustained intracellular pressure gradients. However, these experiments are particularly challenging and with very low throughput. Therefore, we feel that these are out of the scope of the present report and we mention these as promising future directions.

**Recommendations for the authors:**

**Reviewer #1 (Recommendations for the authors):**
Please add more information to Materials and methods and figure captions to more clearly share how many different cells and trials the data are coming from.

This has been done.

Please add the full equation for displacement vs. time for the poroelastic model and describe appropriately.

This cannot be done but we explain why.

Overall, the clarity of the writing in the manuscript could be improved.

This has been done.

Please increase text size in some of the figures.

This has been done.

**Reviewer #2 (Recommendations for the authors):**Figure 1 would benefit from some revisions for clarity. In Panel D, for the control experiment with 7 cells, why are only 3 data points shown?

This was due to the use of excel for generating the box plot. Some data points overlap. We now have used a different software.

In Panel E, there is no legend explaining the red dots in the whisker plots.

This has now been added.

Additionally, the inset in Panel D lacks a legend, and it is unclear how k was computed.

This inset panel has been removed.

Moreover, I find Figure 1, Panel C somewhat pixelated, which makes it challenging to interpret. As I am colorblind, I need to zoom in significantly to distinguish the colors, and the current resolution makes this difficult. Improving the image resolution would be helpful.

Apologies for this. We have now verified the quality of images on our submission.

I am unsure about the method used to compute the relaxation timescale in Figure S2. If an exponential relaxation is assumed, I would expect a function of the form:\begin{document}$$\displaystyle d(t1+tau \_p)= d2+(d 1-d 2)* \exp \left(-(t-t 1) /\left(tau \_p\right)\right)$$\end{document}which implies that for t=t1+tau_p, the result should be d1+0.6*Delta d which does not correspond to the formula given. Have you tried fitting the data with an exponential function or using the model to extract tau_p without assuming a specific functional form?

We thank the reviewer for pointing this out. We have now added further explanation of the fitting to the figure legend.

References:

(1) Rosenbluth, M. J., Crow, A., Shaevitz, J. W. & Fletcher, D. A. Slow stress propagation in adherent cells. Biophys J 95, 6052-6059 (2008). https://doi.org/10.1529/biophysj.108.139139

(2) Esteki, M. H. et al. Poroelastic osmoregulation of living cell volume. iScience 24, 103482 (2021). https://doi.org/10.1016/j.isci.2021.103482

(3) Charras, G. T., Mitchison, T. J. & Mahadevan, L. Animal cell hydraulics. J Cell Sci 122, 3233-3241 (2009). https://doi.org/10.1242/jcs.049262

(4) Moeendarbary, E. et al. The cytoplasm of living cells behaves as a poroelastic material. Nat Mater 12, 253-261 (2013). https://doi.org/10.1038/nmat3517

(5) Luby-Phelps, K., Castle, P. E., Taylor, D. L. & Lanni, F. Hindered diffusion of inert tracer particles in the cytoplasm of mouse 3T3 cells. Proc Natl Acad Sci U S A 84, 4910-4913 (1987). https://doi.org/10.1073/pnas.84.14.4910

(6) Charras, G. T., Coughlin, M., Mitchison, T. J. & Mahadevan, L. Life and times of a cellular bleb. Biophys J 94, 1836-1853 (2008). https://doi.org/10.1529/biophysj.107.113605

(7) Tinevez, J. Y. et al. Role of cortical tension in bleb growth. Proc Natl Acad Sci U S A 106, 18581-18586 (2009). https://doi.org/10.1073/pnas.0903353106